# An Information-theoretic Perspective of Hierarchical Clustering

## Abstract

A combinatorial cost function for hierarchical clustering was introduced by Das-gupta [10]. It has received great attention and several new cost functions from sim-ilar combinatorial perspective have been proposed. In this paper, we investigate hierarchical clustering from the *information-theoretic* perspective and formulate a new objective function. We also establish the relationship between these two perspectives. In algorithmic aspect, we present two algorithms for expander-like and well-clustered cardinality weighted graphs, respectively, and show that both of them achieve $O(1)$-approximation for our new objective function. For practi-cal use, we consider non-binary hierarchical clustering problem. We get rid of the traditional top-down and bottom-up frameworks, and present a new one. Our new framework stratifies the sparsest level of a cluster tree recursively in guide with our objective function. Our algorithm called HCSE outputs a $k$-level cluster tree by an interpretable mechanism to choose $k$ automatically without any hyper-parameter. Our experimental results on synthetic datasets show that HCSE has its own superiority in finding the intrinsic number of hierarchies, and the results on real datasets show that HCSE also achieves competitive costs over the popular non-binary hierarchical clustering algorithms LOUVAIN and HLP.

## 1 Introduction

Hierarchical clustering for graphs plays an important role in the structural analysis of a given data set. Understanding hierarchical structures on the levels of multiple granularities is fundamental in various disciplines including artificial intelligence, physics, biology, sociology, etc [4, 11, 13, 9]. Hierarchical clustering requires a cluster tree that represents a recursive partitioning of a graph into smaller clusters as the tree nodes get deeper. A leaf represents a graph node while a non-leaf node represents a cluster containing its descendant leaves. The root is the largest one containing all leaves.

Clustering is usually formulated as an optimization problem with some objective function. For hier-archical clustering, no cost function with a clear and reasonable combinatorial explanation was de-veloped until Dasgupta [10] introduced a cost function for cluster trees. In this definition, similarity or dissimilarity between data points is represented by weighted edges. Taking the similarity-based metrics as an example, a cluster is a set of nodes with relatively denser intra-links compared with its inter-links, and in a good cluster tree, heavier edges tend to connect leaves whose lowest common ancestor (LCA) is as deep as possible. This intuition leads to Dasgupta's cost function that is a bilinear combination of edge weights and the sizes of corresponding LCAs.

Motivated by Dasgupta's cost function, Cohen-Addad et al. [8] proposed admissible cost functions. In their definition, the size of each LCA in Dasgupta's objective is generalized to be a function of the sizes of its left and right children. For all similarity-based graphs generated from a minimal ultrametric, a cluster tree achieves the minimum cost if and only if it is a generating tree that is a

"natural" ground truth tree in an axiomatic sense therein. A necessary condition of admissibility of an objective function is that it achieves the same value for every cluster tree for a uniformly weighted clique that has no structure in common sense. However, any slight deviation of edge weights would generally separate the two end-points of a light edge on a high level of its optimal (similarity-based) cluster tree. Thus, it seems that admissible objective functions, which take Dasgupta's cost function as a specific form, ought to be an unchallenged criterion in evaluating cluster trees since they are formulated by an axiomatic approach.

However, an admissible cost function seems imperfect in practice. The arbitrariness of optima of cluster trees for cliques indicates that the division of each internal node on an optimal cluster tree totally neglects the *balance* of its two children. Edge weight is the unique factor that decides the structure of optimal trees. But a balanced tree is commonly considered as an ideal candidate in hierarchical clustering compared to an unbalanced one. Even clustering for cliques, a balanced partition should be preferable for each internal node. At least, an optimal cluster tree whose height is logarithm of graph size $n$ is intuitively more reasonable than a caterpillar shaped cluster tree whose height is $n - 1$. Moreover, a simple proof would imply that the optimal cluster tree for any connected graphs is binary. This property is not always useful in practice since a real system usually has its inherent number of hierarchies and a natural partition for each internal cluster. For instance, the natural levels of administrative division in a country is usually intrinsic, and it is not suitable to differentiate hierarchies for parallel cities in the same state. This structure cannot be obtained by simply minimizing admissible cost functions.

In this paper, we investigate the hierarchical clustering from the perspective of information theory. Our study is based on Li and Pan's structural information theory [14] whose core concept named structural entropy measures the complexity of hierarchical networks. We summarize our contributions as follows.

(1) We formulate **a new objective function from the information-theoretic perspective**, which builds the bridge for combinatorial and information-theoretic perspectives for hierarchical clustering. For this cost function, the balance of cluster trees will be involved naturally as a factor just like we design optimal codes, for which the balance of probability over objects is fundamental in constructing an efficient coding tree. We also define cluster trees with a specific height, which is coincident with our cognition of natural clustering.

(2) For our new objective function, we present **two polynomial-time approximation algorithms** respectively for two cases of the conductance $\Phi(G)$ of a cardinality weighted graph $G$. Our first result shows that *any* cluster tree of $G$ has a approximation factor $O(\Phi(G)^{-1})$ (Theorem 3.1). So a "Huffman-merge" heuristic that solely depends on the degrees of vertices achieves this guarantee, and it achieves $O(1)$-approximation when $\Phi(G)$ is a constant. The second result is a $O(1)$-approximation algorithm for $G$ that can be well clustered into a constant number of expanders (Theorem 3.2). The main idea of this algorithm is inspired by very recent Manghiuc and Sun's work [15], and our approximation factors for our new objective also match their results in these two cases.

(3) For practical use, we develop **a new interpretable framework for natural hierarchical clustering** that outputs a non-binary cluster tree. The idea of our framework is essentially different from the traditional recursive division or agglomeration ones. In our framework, the *sparsest level* of the cluster tree is stratified recursively. This coincide with the intuition that when we differentiate the hierarchies of a complex system, the clearest level should be stratified first, rather than in a rigid divisive or agglomerative fashion. Therefore, this framework has much better interpretability than the traditional ones.

(4) We develop **a new non-binary clustering algorithm** (HCSE) under the new clustering framework. To find the sparsest level in each iteration, we formulate two basic operations called *stretch* and *compress*, respectively. HCSE terminates when a specific criterion that intuitively coincides with the natural hierarchies is met, and *no* hyperparameter is needed. Our extensive experiments on both synthetic and real datasets demonstrate that HCSE outperforms the present popular heuristic algorithms LOUVAIN [3] and HLP [19]. These two algorithms proceed simply by recursively invoking flat clustering algorithms based on modularity and label propagation, respectively, and the hierarchy number is solely determined by the number of rounds when the algorithm terminates. So their interpretability is quite poor. Our experimental results on synthetic datasets show that HCSE has a great advantage in finding the intrinsic number of hierarchies, and the results on real datasets show that HCSE achieves much better costs than HLP and competitive costs to LOUVAIN.

**Related work.** The first combinatorial objective function was proposed by Dasgupta [10]. Along with this line of study, several alternative objectives have been presented. All of them are bilinear functions of edge weights and some function of the corresponding LCAs. For Dasgupta's cost function and for the worst case study, Dasgupta showed that a recursively bipartition applying Arora's seminal algorithm for sparsest cut problem [2] yields $O(\log^{1.5} n)$-approximation, and it was improved by [20] and [5, 8] to $O(\log n)$ and $\sqrt{\log n}$, respectively. It is NP-hard to optimize the cluster tree [10] and even a $O(1)$-approximation is impossible under the Small Set Expansion hypothesis [20, 5]. Beyond the worst case, Cohen-Addad et al. [8] showed that a SVD-based algorithm achieves a $O(1 + o(1))$-approximation for the stochastic block model with high probability. Manghiuc and Sun [15] presented a $O(1)$-appromation algorithm for more generalized well-clustered graphs. The outline of their method is to utilize a flat clustering algorithm [12] to obtain the underlying clusters first, and then some relatively easy heuristics for clustering in and out of these clusters are enough for the guarantee. Our proof follows this route also.

For other lines of this study, Moseley and Wang [16] studied the dual of Dasgupta's cost function and showed that the average-linkage algorithm achieves a $(1/3)$-approximation. This factor has been improved by a series of works to $0.336$ [6], $0.4246$ [7] and $0.585$ [1], respectively. Cohen-Addad et al. [8] considered maximization of Dasgupta's cost function for the dissimilarity-based metrics. They proved that the average-link and random partitioning algorithms achieve a $(2/3)$-approximation, which has been improved to $0.667$ [6], $0.716$ [18] and $0.74$ [17], respectively.

For non-binary cluster tree construction, the most popular algorithm for practical use is LOUVAIN [3]. More recently, a hierarchical label propagation based algorithm HLP has been presented [19]. Both of these two algorithms construct a non-binary cluster tree with the same framework, that is, the hierarchies are formed from bottom to top one by one. In each round, they invoke different flat clustering algorithms, Modularity and Label Propagation, respectively.

## 2 A cost function from information-theoretic perspective

In this section, we introduce Li and Pan's structural information theory [14] and the combinatorial cost functions of Dasgupta [10] and Cohen-Addad et al. [8]. Then we propose a new cost function that is developed from structural information theory and establish the relationship between the information-theoretic and combinatorial perspectives.

**Notations.** Let $G = (V, E, w)$ be an undirected weighted graph with a set of vertices $V$, a set of edges $E$ and a weight function $w : E \to \mathbb{R}^+$, where $\mathbb{R}^+$ denotes the set of all positive real numbers. An unweighted multigraph can be viewed as a cardinality weighted one whose edge weight is the number of parallel edges. For each vertex $u \in V$, denote by $d_u = \sum_{(u,v) \in E} w(u,v)$ the weighted degree of $u$. For a subset of vertices $S \subseteq V$, define the volume of $S$ to be the sum of degrees of vertices. We denote it by $\mathrm{vol}(S) = \sum_{u \in S} d_u$. We denote by $G[S]$ the subgraph induced by $S$. A cluster tree $T$ for graph $G$ is a rooted tree with $|V|$ leaves, each of which is labeled by a distinct vertex $v \in V$. Each non-leaf node on $T$ is labeled by a subset $S$ of $V$ that consists of all the leaves treating $S$ as an ancestor. For each node $\alpha$ on $T$, denote by $\alpha^-$ the parent of $\alpha$, and by $|\alpha|$ its size. For each pair of leaves $u$ and $v$, denote by $u \vee v$ the LCA of them on $T$.

**Structural entropy of graphs.** Because of the tense space limit, we just give the definition of the core concept structural entropy in structural information theory. The idea of this definition is briefly introduced in Appendix A. Readers could also refer to [14] for more information on this theory.

Given a weighted graph $G = (V, E, w)$ and a cluster tree $T$ for $G$, the *structural entropy of $G$ on $T$* is defined as

$$\mathcal{H}^T(G) = -\sum_{\alpha \in T} \frac{g_\alpha}{\mathrm{vol}(V)} \log \frac{\mathrm{vol}(\alpha)}{\mathrm{vol}(\alpha^-)},^1$$

where $\alpha^-$ denotes the parent of tree node $\alpha$, and $g_\alpha$ denotes the sum of weights of edges in $G$ with exactly one end-point in the set of vertices corresponding to $\alpha$. The *structural entropy of $G$* is defined as the minimum one among all cluster trees, denoted by $\mathcal{H}(G) = \min_T \{\mathcal{H}^T(G)\}$.

---

[1] For notational convenience, for the root $\lambda$ of $T$, set $\lambda^- = \lambda$. So the term for $\lambda$ in the summation is 0. In this paper, the omitted base of logarithm is always 2.

**Combinatorial explanation of structural entropy.** The cost function of a cluster tree $T$ for graph $G = (V, E)$ introduced by Dasgupta [10] is defined to be $c^T(G) = \sum_{(u,v) \in E} w(u,v)|u \vee v|$. The admissible cost function introduced by Cohen-Addad et al. [8] generalizes the term $|u \vee v|$ in the definition of $c^T(G)$ to be a general function $g(|L|, |R|)$, where $L$ and $R$ are the two children of $u \vee v$, respectively. Dasgupta defined $g(x, y) = x + y$. For both definitions, the optimal hierarchical clustering of $G$ is in correspondence with a cluster tree of minimum cost in the combinatorial sense that heavy edges are cut as far down the tree as possible. The following proposition establishes the relationship between structural entropy and this kind of combinatorial form of cost functions.

**Proposition 2.1.** *For a weighted graph $G = (V, E, w)$, minimizing $\mathcal{H}^T(G)$ (over $T$) is equivalent to minimizing the cost function*

$$cost^T(G) = \sum_{(u,v) \in E} w(u,v) \log vol(u \vee v). \tag{1}$$

We defer the proof of Proposition 2.1 to Appendix B. We call cost(SE) the cost function in Proposition 2.1 from now on. Proposition 2.1 indicates that when we view $g$ as a function of the LCA rather than that of its size and define $g(u, v) = \log vol(u \vee v)$, the "admissible" function becomes equivalent to structural entropy in evaluating cluster trees, although it is not admissible any more.

So what is the difference between these two cost functions? As stated by Cohen-Addad et al. [8], an important axiomatic hypothesis for admissible function, thus also for Dasgupta's cost function, is that the cost for every binary cluster tree of an unweighted clique is identical. So any binary tree for clustering on cliques is reasonable, which coincides with the common sense that structureless datasets can be organized hierarchically free. However, for structural entropy, the following theorem indicates that balanced organization is of importance even though for structureless dataset.

**Proposition 2.2.** *For any positive integer $n$, let $K_n$ be the clique of $n$ vertices with identical weight on every edge. Then a cluster tree $T$ of $K_n$ achieves minimum structural entropy if and only if $T$ is a balanced binary tree, that is, the two children clusters of each sub-tree of $T$ have difference in size at most $1$.*

The proof of Proposition 2.2 is a bit technical, and we defer it to Appendix C. The intuition behind Proposition 2.2 is that balanced codes are the most efficient encoding scheme for unrelated data. So the codewords of the random walk that jumps freely among clusters on each level of a cluster tree have the minimum average length if all the clusters on this level are in balance.

It is worth noting that the admissible function introduced by Cohen-Addad et al. [8] is defined from the viewpoint that a generating tree $T$ of a similarity-based graph $G$ that is generated from a minimal ultrametric achieves the minimum cost. In this definition, the monotonicity of edge weights between clusters on each level from bottom to top on $T$, which is given by Cohen-Addad et al. [8] as a property of a "natural" ground-truth hierarchical clustering, is the unique factor when evaluating $T$. However, Proposition 2.2 implies that for cost(SE), besides cluster weights, the balance of cluster trees is implicitly involved as another factor. Moreover, for cliques, the minimum cost should be achieved on every subtree, which makes an optimal cluster tree balanced everywhere. This optimal clustering for cliques is also robust in the sense that a slight perturbation to the minimal ultrametric, which can be considered as slight variations to the weights of a batch of edges, will not change the optimal cluster tree structure wildly due to the holdback force of balance.

## 3 Approximation algorithms for SE-based cost function

In this section, we present approximation algorithms for expander-like and well-clustered graphs, respectively. These algorithms work for cardinality edge weights (e.g. the multiplicity of edges).

**Why cardinality weights?** In general, the term $\log vol(u \vee v)$ in Eq. 1 and thus cost(SE) may be negative when the volume of $u \vee v$ varies, which may lead to pathosis in approximation analysis. The cardinality weight function $w$ is at least one, which makes cost(SE) non-negative. The dependence of cost(SE) on the scale of edge weights violates the scale-invariance principle. However, we emphasize that $\mathcal{H}^T(G)$ is scale-invariant and Proposition 2.1 holds for any scale variation. In this paper, we present approximation algorithms for cost(SE) in well-defined settings.

**Theorem 3.1.** *For any cardinality weighted graph $G = (V, E, w)$ with conductance $\Phi(G)$, it holds that any cluster tree has a cost $O(\Phi(G)^{-1}) \cdot OPT$, where $OPT$ is the minimum cost(SE) of $G$.*

We defer the proof of Theorem 3.1 to Appendix D. When $\Phi(G)$ is a constant, Theorem 3.1 implies that any cluster tree achieves $O(1)$-approximation for expanders. Thus, the balance of a cluster tree has a significant impact on its cost. Considering balance as an important factor, we present a Huffman-merging heuristic (Algorithm 1). It will serve as a subroutine for the algorithm for well-clustered graphs.

---

**Algorithm 1:** HuffmanMerge

---

**Input:** a graph $G = (V, E, w)$
**Output:** A cluster tree $T$ of $G$
1 Create $n$ singleton trees;
2 **while** *there are at least two trees* **do**
3     Select the two trees $T_1$ and $T_2$ with the least volumes;
4     Construct a new tree $T_0$ with $T_1$ and $T_2$ as two subtrees of the root;

5 **Return** the resulting binary tree $T_0$.

---

Next, we consider well-clustered graphs that are composed by a collection of densely-connected components with high inner conductance and weakly interconnections. Our settings for well-clustered graphs is the same as those in [15]. We start from the following $(\Phi_{in}, \Phi_{out})$-decomposition presented by Gharan and Trevisan [12]. Let $\lambda_k$ be the $k$-th smallest eigenvalue of the normalized Laplacian matrix of $G$ and $\Phi_G(S)$ be the conductance of a vertex set $S$ in $G$.

**Lemma 3.1.** *([12], Theorem 1.5) Let $G = (V, E, w)$ be a graph such that $\lambda_k > 0$, for some $k \geq 1$. Then, there is a local search algorithm that finds a $l$-partition $\{P_i\}_{i=1}^l$ of V, for some $l < k$, such that for every $1 \leq i \leq l$, $\Phi_G(P_i) = \mathcal{O}(k^6\sqrt{\lambda_{k-1}})$ and $\Phi(G[P_i]) = \Omega(\lambda_k^2/k^4)$.*

Lemma 3.1 implies that, when graph $G$ exhibits a clear clustering structure, there is a partition $\{P_i\}_{i=1}^l$ of V such that for each $P_i$ both the outer and inner conductance can be bounded. This is one of the most crucial insights that we can use $\{P_i\}_{i=1}^l$ directly to construct a cluster tree.

For a high-level description, our algorithm consists of two phases: Partition and Merge. In the Partition phase, it invokes the algorithm in Lemma 3.1 to partion $V$ into sets $\{P_i\}_{i=1}^l$. In the Merge phase it combines the trees in a "caterpillar style" according to an increasing order of their volumes. This algorithm is described as Algorithm 2.

---

**Algorithm 2:** CaterpillarMerge

---

**Input:** A graph $G = (V, E, w)$, an integer $k \geq 2$ such that $\lambda_k > 0$
**Output:** A cluster tree $T$ of $G$
1 Apply the partitioning algorithm in Lemma 3.1 on input $(G, k)$ to obtain $\{P_i\}_{i=1}^l$ for some $l < k$;
2 Sort $P_1, ..., P_l$ be such that $\mathrm{vol}_G(P_i) \leq \mathrm{vol}_G(P_{i+1})$, for all $1 \leq i < l$;
3 Let $T_i = \mathrm{HuffmanMerge}(G[P_i])$;
4 Initialize $T = T_1$;
5 **for** $i = 2, ..., l$ **do**
6     Let $T$ be the tree with $T$ and $T_i$ as its two children;

7 **Return** $T$.

---

Note that Algorithm 2 degenerates to Algorithm 1 when $k = 2$. For the approximation guarantee, we have the following theorem.

**Theorem 3.2.** *Let $G = (V, E, w)$ be a cardinality weighted graph such that $\lambda_k > 0$ for some $k \geq 1$. Then Algorithm 2 constructs in polynomial time a cluster tree $T$ of $G$ that achieves $O\left(\frac{1}{(1-\alpha)\beta} \log \frac{k}{1-\alpha}\right)$-approximation for $cost^T(G)$, where $\alpha = O(k^6\sqrt{\lambda_{k-1}})$, $\beta = \Omega(\lambda_k^2/k^4)$. Consequently, when $\lambda_k = \Omega(1/poly(k))$ and $\lambda_{k-1} = O(1/k^{12})$ such that $\alpha < 1 - \rho$ for some constant $\rho \in (0, 1)$, Algorithm 2 achieves $O(poly(k))$-approximation. In addition, when $k$ is a constant, Algorithm 2 achieves $O(1)$-approximation.*

The proof of Theorem 3.2 is given in Appendix E.

# 4 Practically used non-binary hierarchical clustering algorithm

In this section, we develop a non-binary hierarchical clustering algorithm based on cost(SE) optimization. At present, all existing algorithms for hierarchical clustering can be categorized into two frameworks: top-down division and bottom-up agglomeration [8]. The top-down division approach usually yields a binary tree by recursively dividing a cluster into two parts with a cut-related criterion. But a binary clustering tree is far from a practical one as we introduced in Section 1. For practical use, bottom-up agglomeration that is also known as hierarchical agglomerative clustering (HAC) is commonly preferable. It constructs a cluster tree from leaves to the root recursively, during each round of which the newly generated clusters shrink into single vertices.

Our algorithm jumps out of these two frameworks. We establish a new one that stratifies the *sparsest* level of a cluster tree recursively rather than in a sequential order. In general, in guide with cost(SE), we construct a $k + 1$-level cluster tree from the previous $k$-level one, during which we find the level whose stratification makes the average cost in a local reduced subgraph decrease most, and then differentiate it into two levels. The process of stratification consists of two basic operations: *stretch* and *compression*. Generally speaking, in stretch steps, given an internal node of a cluster tree, a local binary subtree is constructed, while in compression steps, the paths that are overlength from the root to leaves on the binary tree is compressed by shrinking tree edges that make the cost reduce most. The intuition behind the "stretch-and-compress" scheme is as follows. First, we run a fast and simple, but probably rough clustering algorithm to obtain a binary cluster subtree. So intuitively, after stretch, we unfold all the potential hierarchies such that the sparsest level is possibly to be seen. Second, we compress every overlength path that is supposed to get through each level of this subtree, during which, the edge on the sparsest level whose compression makes too many graph edges amplify the sizes of their LCAs to a large extent will be retained.

We remark that this framework can be collocated with any cost function and any binary cluster tree algorithm. For computational efficiency, especially for real networks of large scale more than $10^4$, we will adopt in our experiments an HAC construction of binary cluster trees in stretch steps.

**Stretch and compress.** Given a cluster tree $T$ for graph $G = (V, E)$, let $u$ be an internal node on $T$ and $v_1, v_2, \ldots, v_\ell$ be its children. We call this local parent-children structure rooted at $u$ to be a $u$-*triangle* of $T$, denoted by $T_u$. These two operations are defined on $u$-triangles. Note that each child $v_i$ of $u$ is a cluster in $G$. We reduce $G$ by shrinking each $v_i$ to be a single vertex $v_i'$ while maintaining each inter-link and ignoring each internal edge of $v_i$. This reduction captures the connections of clusters at this level in the parent cluster $u$. The stretch operation constructs a binary tree for $u$-triangle. We adopt a common HAC construction in this $u$-triangle. That is, initially, view each $v_i'$ as a cluster and recursively combine two clusters into a new one for which cost(SE) drops most. The sequence of combinations yields a binary subtree $T_u'$ rooted at $u$ which has $v_1, v_2, \ldots, v_\ell$ as leaves. Then the compression operation is proposed to reduce the height of $T_u'$ to be 2. Let $\hat{E}(T')$ be the set of edges on $T'$, each of which appears on a path of length more than 2 from the root of $T'$ to some leaf. Denote by $\Delta(e)$ for edge $e$ be the amount of structural entropy enhanced by the shrink of $e$. We pick from $\hat{E}(T_u')$ the edge $e$ with least $\Delta(e)$. Note that the compression of a tree edge makes the grandchildren of some internal node to be children, which must amplify the cost. The compression operation picks the least amplification. The processes of stretch and compress are illustrated in Figure 3 and stated in Algorithms 5 and 6, respectively (see Appendix G).

**Sparsest level.** Let $U_j$ be the set of $j$-level nodes on cluster tree $T$, that is, $U_j$ is the set of nodes each of which has distance $j$ from $T$'s root. Suppose that the height of $T$ is $k$, then $U_0, U_1, \ldots, U_{k-1}$ is a partition for all internal nodes of $T$. For each internal node $u$, define $\mathcal{H}(u) = -\sum_{v: v^- = u} \frac{g_u}{\text{vol}(V)} \log \frac{\text{vol}(v)}{\text{vol}(u)}$. Note that $\mathcal{H}(u)$ is the partial sum contributed by $u$ in $\mathcal{H}^T(G)$. After a "stretch-and-compress" round on $u$-triangle, denote by $\Delta\mathcal{H}(u)$ the structural entropy by which the new cluster tree reduces. Since the reconstruction of $u$-triangle stratifies cluster $u$, $\Delta\mathcal{H}(u)$ is always non-negative. Define the sparsity of $u$ to be $\text{Spar}(u) = \frac{\Delta\mathcal{H}(u)}{\mathcal{H}(u)}$, which is the relative variation of structural entropy in cluster $u$. From the information-theoretic perspective, this means that the uncertainty of random walk can be measured locally in any internal cluster, which reflects the quality of clustering in this local area. At last, we define the *sparsest level* of $T$ to be the $j$-th level such that the average sparsity of triangles rooted at nodes in $U_j$ is maximum, that is $\arg\max_j\{\overline{\text{Spar}}_j(T)\}$,

where $\overline{\mathrm{Spar}}_j(T) = \sum_{u \in U_j} \mathrm{Spar}(u)/|U_j|$. Then stratification works for the sparsest level of $T$. This process is illustrated in Figure 4 (see Appendix G).

For a given positive integer $k$, to construct a cluster tree of height $k$, we start from the trivial 1-level cluster tree that involves all vertices of $G$ as leaves. Then we do not stop stratifying at the sparsest level recursively until a $k$-level cluster tree is obtained. This process is described in Algorithm 3.

---

**Algorithm 3:** $k$-Hierarchical clustering based on structural entropy ($k$-HCSE)

**Input:** a graph $G = (V, E)$, $k \in \mathbb{Z}^+$
**Output:** a $k$-level cluster tree $T$
1 Initialize $T$ to be the 1-level cluster tree;
2 $h = \mathrm{height(T)}$;
3 **while** $h < k$ **do**
4 $\quad$ $j' \leftarrow \arg\max_j \{\overline{\mathrm{Spar}}_j(T)\}$; $\quad$ // Find the sparsest level of $T$ (breaking ties arbitraily);
5 $\quad$ **if** $\overline{Spar}_{j'}(T) = 0$ **then**
6 $\quad\quad$ break; $\quad$ // No cost will be saved by any further clustering;
7 $\quad$ **for** $u \in U_{j'}$ **do**
8 $\quad\quad$ $T_u \leftarrow \mathrm{Stretch}(u\text{-triangle } T_u)$;
9 $\quad\quad$ $\mathrm{Compress}(T_u)$;
10 $\quad$ $h \leftarrow h + 1$;
11 $\quad$ **for** $j \in [j'+1, h]$ **do**
12 $\quad\quad$ Update $U_j$;
13 **return** $T$

---

To determine the height of the cluster tree automatically, we derive the natural clustering from the variation of sparsity on each level. Intuitively, a natural hierarchical cluster tree $T$ should have not only sparse boundary on clusters, but also low sparsity for triangles of $T$, which means that stratification within the reduced subgraphs corresponding to the triangles on the sparsest level makes little sense. For this reason, we consider the inflection points of the sequence $\{\delta_t(\mathcal{H})\}_{t=1,2,\ldots}$, where $\delta_t(\mathcal{H})$ is the structural entropy by which the $t$-th round of stratification reduces. Formally, denote $\Delta_t \mathcal{H} = \delta_{t-1}(\mathcal{H}) - \delta_t(\mathcal{H})$ for each $t \geq 2$. We say that $\Delta_t \mathcal{H}$ is an inflection point if both $\Delta_t \mathcal{H} \geq \Delta_{t-1} \mathcal{H}$ and $\Delta_t \mathcal{H} \geq \Delta_{t+1} \mathcal{H}$ hold. Our algorithm finds the least $t$ such that $\Delta_t \mathcal{H}$ is an inflection point and fix the height of the cluster tree to be $t$ (Note that after $t-1$ rounds of stratification, the number of levels is $t$). This process is described as Algorithm 4.

---

**Algorithm 4:** Hierarchical clustering based on structural entropy (HCSE)

**Input:** a graph $G = (V, E)$
**Output:** a cluster tree $T$
1 $t \leftarrow 2$;
2 **while** $\Delta_t \mathcal{H} < \Delta_{t-1} \mathcal{H}$ *or* $\Delta_t \mathcal{H} < \Delta_{t+1} \mathcal{H}$ **do**
3 $\quad$ **if** $\max_j \{\overline{Spar}_j(T)\} = 0$ **then**
4 $\quad\quad$ break;
5 $\quad$ $t \leftarrow t + 1$;
6 **return** $t$-HCSE$(T)$

---

**Time complexity.** The running time of HCSE on graph $G = (V, E)$ for which $|V| = n$ and $|E| = m$ depends mainly on the iterations of stratification for the sparsest level. For each round of $t$-HCSE in Algorithm 4, since the change of structure entropy can be calculated incrementally and locally when merge siblings, the time complexity for the Stretch process is $O(mh \log n)$, where $h$ is the height of the binary tree that Stretch yields. Since at most $n$ times of shrinking operations on tree edges will happen, the time complexity for the Compress process is $O(hn)$. Let $h_{\max}$ be the maximum height among the binary trees that appear during all iterations. The time complexity of HCSE (and also $k$-HCSE) is $O(kmh_{\max} \log n + kh_{\max}n)$. In practice, $k$ is usually very small (we can even set $k = O(1)$ in $k$-HCSE). Moreover, the balance property of structural entropy tends to produce a

| | $\vec{p}$ | HCSE | HLP | LOU |
|---|---|---|---|---|
| $p_2$ | 4.5E(-2) | 0.89 | 0.79 | **0.92** |
| $p_1$ | 1.5E(-3) | **0.93** | 0.75 | 0.92 |
| $p_0$ | 6E(-6) | **0.62** | 0.58 | $--$ |
| $p_2$ | 5.5E(-2) | 0.87 | **0.89** | 0.89 |
| $p_1$ | 1.5E(-3) | **0.95** | 0.87 | 0.87 |
| $p_0$ | 4E(-6) | **0.72** | $--$ | $--$ |
| $p_2$ | 6.5E(-2) | 0.96 | 0.95 | **0.99** |
| $p_1$ | 4.5E(-3) | 0.94 | 0.81 | **0.99** |
| $p_0$ | 2.5E(-6) | **0.80** | $--$ | $--$ |

Table 1: NMI for three algorithms. Each dataset has $2,500$ vertices, and the cluster numbers at three levels are 5, 25 and 250, respectively, for which the size of each cluster is accordingly generated at random. $p_3 = 0.9$ for each graph. "$--$" means the algorithm does not find this level.

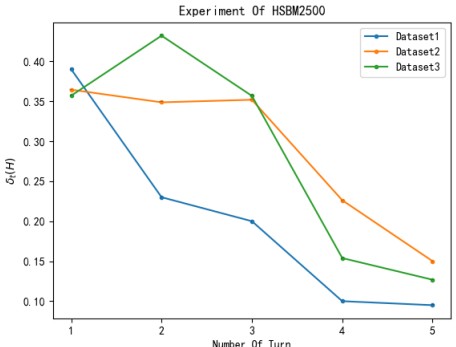

Figure 1: $\delta_t(\mathcal{H})$ variations for HCSE. It can be observed easily that the inflection points for all the three datasets appear on $t = 4$, which is also the ground-truth number of hierarchies.

balanced binary tree after stretch, which makes $h_{\max} = O(\log n)$. Therefore, in this case, the time complexity is merely $O(m \log^2 n)$.

# 5 Experiments

We evaluate experimentally our practically used non-binary clustering algorithm both on synthetic networks generated from the Hierarchical Stochastic Block Model (HSBM) and on real datasets, respectively. We compare HCSE with the popular practical algorithms LOUVAIN [3] and HLP [19]. To avoid over-fitting to higher levels, which possibly results in under-fitting to lower levels, LOUVAIN admits a sequential input of vertices. Usually, to avert the worst-case trap, the vertices come randomly, and the resulting cluster tree depends on their order. HLP invokes the common LP algorithm recursively, and so it cannot be guaranteed to avoid under-fitting in each round. This can be seen in our experiments on synthetic datasets, for which these two algorithms usually miss ground-truth levels. For real datasets, as far as we know, no public real datasets have clear ground truth for hierarchical clustering. We do the comparative experiments on real networks. Some of them have (overlapping, possibly hierarchical) ground truth, e.g., Amazon, while others do not have. We evaluate the resulting cluster trees for the Amazon network by Jaccard index, and show the results in Appendix F, For other networks without ground truth, we evaluate results by both cost(SE) and Dasgupta's cost function cost(Das). All the source code can be downloaded from https://github.com/samwu-learn/HCSE.

**Synthetic datasets generated from HSBM.** Our experiments on synthetic datasets utilize 4-level HSBM. For simplicity, let $\vec{p} = (p_0, p_1, p_2, p_3)$ be the probability vector for which $p_i$ is the probability of generating edges for vertex pairs whose LCA on the ground-truth cluster tree has depth $i$. Note that the 0-depth node is the root. We compare the Normalized Mutual Information (NMI) at each level of the ground-truth cluster tree to those of three algorithms. Note that the randomness in LOUVAIN, and breaking-ties rule as well as convergence of HLP make different results, we choose the most effective strategy and pick the best results in five runs for both of them. Compared to their uncertainty, our algorithm HCSE yields stable results.

Table 1 demonstrates the results in three groups of probabilities, for which the hierarchical structures get clearer one by one. Each dataset has $2,500$ vertices, and the cluster numbers at three levels are 5, 25 and 250, respectively, for which the size of each cluster is accordingly generated at random. $p_3 = 0.9$ for each graph. Our algorithm HCSE is always able to find the right number of levels, while LOUVAIN always misses the top level, and HLP misses the top level in two groups. The inflection points for choosing the intrinsic hierarchy number $t = 4$ of hierarchies are demonstrated in Figure 1.

| Networks | HCSE | HLP | LOUVAIN |
|---|---|---|---|
| CSphd | 1.30E4 / **5.19E4** / 5 | 1.54E4 / 5.58E4 / 4 | **1.28E4** / 7.61E4 / 5 |
| fb-pages-government | 2.48E6 / **1.18E8** / 4 | 2.53E6 / 1.76E8 / 3 | **2.43E6** / 1.33E8 / 4 |
| email-univ | 1.16E5 / **2.20E6** / 3 | 1.46E5 / 6.14E6 / 3 | **1.14E5** / 2.20E6 / 4 |
| fb-messages | 1.58E5 / **4.50E6** / 4 | 1.76E5 / 8.12E6 / 3 | **1.52E5** / 4.96E6 / 4 |
| G22 | **5.56E5** / **2.68E7** / 4 | 6.11E5 / 4.00E7 / 3 | 5.63E5 / 2.80E7 / 5 |
| As20000102 | 2.64E5 / **2.36E7** / 4 | 3.62E5 / 7.63E7 / 3 | **2.42E5** / 2.42E7 / 5 |
| bibd-13-6 | **7.41E5** / **2.56E7** / 3 | 8.05E5 / 4.41E7 / 2 | 7.50E5 / 2.75E7 / 4 |
| delaunay-n10 | 4.65E4 / **3.39E5** / 4 | 4.87E4 / 3.55E5 / 4 | **4.24E4** / 4.25E5 / 5 |
| p2p-Gnutella05 | 9.00E5 / **1.48E8** / 3 | 1.01E6 / 2.78E8 / 3 | **8.05E5** / 1.49E8 / 5 |
| p2p-Gnutella08 | 5.59E5 / **5.51E7** / 4 | 6.36E5 / 1.28E8 / 4 | **4.88E5** / 6.03E7 / 5 |

Table 2: "cost(SE) / cost(Das) / $k$" for three algorithms, where $k$ is the hierarchy number that the algorithm finds.

**Real datasets.** We do our experiments on a series of real networks [2] without ground truth. We compare cost(SE) and cost(Das), respectively. Since the different level numbers given by the three algorithms influence the costs seriously, that is, lower costs are obtained just due to greater heights, we only list in Table 2 the networks for which the three algorithms yield similar level numbers that differ by at most 1 or 2. It can be observed that HLP does not achieve optima for any network, while HCSE performs best w.r.t. cost(Das) for all networks, but does not outperform LOUVAIN for most networks. This is mainly due to the fact that LOUVAIN always finds no less number of hierarchies than HCSE, and the better cost benefits from its depth. Moreover, we emphasize that there is no evidence to indicate that the lower cost(SE) or cost(Das) is, the better a non-binary cluster tree is. Our experiments on these real datasets are just demonstrations of the effectiveness for our interpretable mechanism in hierarchical clustering.

## 6 Conclusions and future discussions

In this paper, we investigate the hierarchical clustering problem from an information-theoretic perspective and propose a new objective function that relates to the combinatorial cost functions raised by Dasgupta [10]. For optimization of this function, we present two $O(1)$-approximation algorithms for expander-like and well-clustered cardinality weighted graphs, respectively. For practical use, we propose a new interpretable non-binary hierarchical clustering framework that stratifies the sparsest level of the cluster tree recursively, which can be collocated with any cost function. We also present an interpretable strategy to find the intrinsic number of levels without any hyper-parameter. The experimental results on $k$-level HSBM demonstrate that our algorithm HCSE has a great advantage in finding $k$ compared to the popular but strongly heuristic algorithms LOUVAIN and HLP. Our results on real datasets show that HCSE also achieves competitive costs compared to these two algorithms.

There are several directions that are worth further study. The first problem is about the relationship between the concavity of $g$ of the cost function and the balance of the optimal cluster tree. It can be checked that for cliques, being concave is not a sufficient condition for total balance. Whether is it a necessary condition? Moreover, is there any explicit necessary and sufficient condition for total balance of the optimal cluster tree for cliques? The second problem is about approximation algorithms for both structural entropy and cost(SE) in the worst case. Due to the non-linear and volume-related function $g$, many previous proof techniques for approximation algorithms seems unavailable. The third one is about more precise characterizations for "natural" hierarchical clustering whose depth is limited. Since any reasonable choice of $g$ makes the cost function achieve optimum on some binary tree, a blind pursuit of minimization of cost functions seems not to be a rational approach. More criteria in this scenario need to be studied.

---

[2]http://networkrepository.com/index.php

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

# A   A brief introduction to structural information

The idea of structural information is to encode a random walk with a certain rule by using a high-dimensional encoding system for a graph $G$. It is well known that a random walk, for which a neighbor is randomly chosen with probability proportional to edge weights, has a stationary distribution on vertices that is proportional to vertex degree.[3] So to position a random walk under its stationary distribution, the amount of information needed is typically the Shannon's entropy, denoted by

$$\mathcal{H}^{(1)}(G) = -\sum_{v \in V} \frac{d_v}{\text{vol}(V)} \log \frac{d_v}{\text{vol}(V)}.$$

By Shannon's noiseless coding theorem, $\mathcal{H}^{(1)}(G)$ is the limit of average code length generated from the *memoryless* source for one step of the random walk. However, dependence of locations may shorten the code length. For each level on cluster trees, the uncertainty of locations is measured by the entropy of the stationary distribution on the clusters of this level. Consider an encoding for every cluster, including the leaves. Each non-root node $\alpha$ is labeled by its order among the children of its parent $\alpha^-$. So the amount of self-information of $\alpha$ within this local parent-children substructure is $-\log(\text{vol}(\alpha)/\text{vol}(\alpha^-))$, which is also roughly the length of Shannon code for $\alpha$ and its siblings. The codeword of $\alpha$ consists of the sequential labels of nodes along the unique path from the root (excluded) to itself (included). The key idea is as follows. For one step of the random walk from $u$ to $v$ in $G$, to indicate $v$, we omit from $v$'s codeword the longest common prefix of $u$ and $v$ that is exactly the codeword of $u \vee v$. This means that the random walk takes this step in the cluster $u \vee v$ (and also in $u \vee v$'s ancestors) and the uncertainty at this level may not be involved. Therefore, intuitively, a quality similarity-based cluster tree would trap the random walk with high frequency in the deep clusters that are far from the root, and long codeword of $u \vee v$ would be omitted. This shortens the average code length of the random walk. Note that we ignore the uniqueness of decoding since a practical design of codewords is not our purpose. We utilize this scheme to evaluate and differentiate hierarchical structures.

Then we formulate the above scheme and measure the average code length as follows. Given a weighted graph $G = (V, E, w)$ and a cluster tree $T$ for $G$, note that under the stationary distribution, the random walk takes one step out of a cluster $\alpha$ on $T$ with probability $g_\alpha/\text{vol}(V)$. Therefore, the aforementioned uncertainty measured by the average code length is

$$\mathcal{H}^T(G) = -\sum_{\alpha \in T} \frac{g_\alpha}{\text{vol}(V)} \log \frac{\text{vol}(\alpha)}{\text{vol}(\alpha^-)},$$

which is defined as the structural entropy of $G$ on $T$. To minimize this uncertainty, the structural entropy $\mathcal{H}(G)$ of $G$ is defined as the minimum one among all cluster trees. Note that the structural entropy of $G$ on the trivial 1-level cluster tree is consistent with the previously defined $\mathcal{H}^{(1)}(G)$. It doesn't have any non-trivial cluster.

---

[3]For connected graphs, this stationary distribution is unique, but not for disconnected ones. Here, we consider this canonical one for all graphs.

 # B  Proof of Proposition 2.1

*Proof.* For each internal node $\alpha$ on $T$, denote by $\partial(\alpha)$ the sets of edges in $G$ with exactly one end-point in the set of vertices corresponding to $\alpha$. So $g_\alpha = \sum_{e \in \partial(\alpha)} w(e)$. Note that

$$
\begin{aligned}
\mathcal{H}^T(G) & = -\sum_{\alpha \in T} \frac{g_\alpha}{\text{vol}(V)} \log \frac{\text{vol}(\alpha)}{\text{vol}(\alpha^-)} \\
& = -\sum_{\alpha \in T} \sum_{(u,v) \in \partial(\alpha)} \frac{w(u,v)}{\text{vol}(V)} \log \frac{\text{vol}(\alpha)}{\text{vol}(\alpha^-)} \\
& = -\sum_{(u,v) \in E} \left( \frac{w(u,v)}{\text{vol}(V)} \sum_{\alpha:(u,v) \in g_\alpha} \log \frac{\text{vol}(\alpha)}{\text{vol}(\alpha^-)} \right).
\end{aligned}
$$

For a single edge $(u,v) \in E$, all the terms $\log(\text{vol}(\alpha)/\text{vol}(\alpha^-))$ for leaf $u$ satisfying $(u,v) \in g_\alpha$ sum (over $\alpha$) up to $\log(d_u/\text{vol}(u \vee v))$ along the unique path from $u$ to $u \vee v$. It is symmetric for $v$. Therefore, considering ordered pair $(u,v) \in E$,

$$
\begin{aligned}
\mathcal{H}^T(G) & = -\sum_{\text{ordered } (u,v) \in E} \frac{w(u,v)}{\text{vol}(V)} \log \frac{d_u}{\text{vol}(u \vee v)} \\
& = \frac{1}{\text{vol}(V)} \left( -\sum_{u \in V} d_u \log d_u + \sum_{\text{ordered } (u,v) \in E} w(u,v) \log \text{vol}(u \vee v) \right) \\
& = \frac{1}{\text{vol}(V)} \left( -\sum_{u \in V} d_u \log d_u + 2 \cdot \sum_{(u,v) \in E} w(u,v) \log \text{vol}(u \vee v) \right).
\end{aligned}
$$

The second equality follows from the fact $\sum_{u \in V} d_u = \sum_{\text{ordered } (u,v) \in E} w(u,v) = \text{vol}(V)$ and the last equality from the symmetry of $(u,v)$. Since the first summation is independent of $T$, Proposition 2.1 follows. $\qquad\square$

# C  Proof of Proposition 2.2

We restate Proposition 2.2 as follows.

**Theorem 2.2.** *For any positive integer $n$, let $K_n$ be the clique of $n$ vertices with identical weight on every edge. Then a cluster tree $T$ of $K_n$ achieves minimum structural entropy if and only if $T$ is a balanced binary tree, that is, the two children clusters of each sub-tree of $T$ have difference in size at most $1$.*

Note that a balanced binary tree (BBT for abbreviation) means the tree is balanced on every internal node. Formally, for an internal node of cluster size $k$, its two sub-trees are of cluster sizes $\lfloor k/2 \rfloor$ and $\lceil k/2 \rceil$, respectively.

For cliques, since the weights of each edge are identical, we assume it safely to be $1$. By Theorem 2.1, minimizing the structural entropy is equivalent to minimizing the cost function (over $T$)

$$
\begin{aligned}
\text{cost}^T(G) & = \sum_{(u,v) \in E} \log \text{vol}(u \vee v) \\
& = \sum_{(u,v) \in E} \log \left( (n-1)|u \vee v| \right) \\
& = \sum_{(u,v) \in E} \log(n-1) + \sum_{(u,v) \in E} \log |u \vee v|
\end{aligned}
$$

Since the first term in the last equation is independent of $T$, the optimization turns to minimizing the last term, which we denote by $\Gamma(T)$. Grouping all edges in $E$ by LCA of two end-points, the cost $\Gamma(T)$ can be written as the sum of the cost $\gamma$ at every internal node $N$ of $T$. Formally, for every

internal node $N$, let $A, B \subseteq V$ be the leaves of the sub-trees rooted at the left and right child of $N$, respectively. We have

$$\Gamma(T) \;=\; \sum_N \gamma(N)$$

$$\gamma(N) \;=\; \left( \sum_{x \in A, y \in B} 1 \right) \cdot \log \left( |A| + |B| \right)$$

$$=\; |A| \cdot |B| \cdot \log(|A| + |B|)$$

Now we only have to show the following lemma.

**Lemma C.1.** *For any positive integer $n$, a cluster tree $T$ of $K_n$ achieves minimum cost $\Gamma(T)$ if and only if $T$ is a BBT.*

*Proof.* Lemma C.1 is proved by induction on $|V|$. The key technique of tree swapping we use here is inspired by Cohen-Addad et al [4]. The basis step holds since for $|V| = 2$ or $3$, the cluster tree is balanced and unique. It certainly achieves the minimum cost exclusively.

Now, consider a clique $G = (V, E)$ with $n = |V| \geq 4$. Let $T_1$ be an arbitrary unbalanced cluster tree and $\lambda$ be its root. We need to prove that the cost $\Gamma(T_1)$ does not achieve the minimum. Without loss of generality, we can safely assume the root node is unbalanced, since otherwise, we set $T_1$ to be the sub-tree that is rooted at an unbalanced node. Let $T_2$ be a tree with root $\lambda$ whose left and right sub-trees are BBTs such that they have the same sizes with the left and right sub-trees of $T_1$, respectively. Let $V_{ll}, V_{lr}, V_{rl}$ and $V_{rr}$ be the sets of nodes on the four sub-trees at the second level of $T_2$ and $n_{ll}, n_{lr}, n_{rl}$ and $n_{rr}$ denote their sizes, respectively. Our proof is also available when some of them are empty. We always assume $n_{ll} \leq n_{lr}$ and $n_{rl} \geq n_{rr}$. Next, we construct $T_3$ by swapping (transplanting) $V_{lr}$ and $V_{rl}$ with each other. Finally, let $T_4$ be a tree with root $\lambda$ whose left and right sub-trees are BBTs after balancing the left and right sub-trees of $T_3$. So $T_4$ is a BBT. Then we only have to prove that $\Gamma(T_1) > \Gamma(T_4)$. Note that the strict ">" is necessary since we need to negate all unbalanced cluster trees.

Then we show that the transformation process that consists of the above three steps makes the cost decrease step by step. Formally,

     (a) $T_1$ to $T_2$. The sub-trees of $T_1$ become BBTs in $T_2$. Since the number of edges whose end-points treat the root as LCA is the same, by induction we have $\Gamma(T_1) \geq \Gamma(T_2)$.

     (b) $T_2$ to $T_3$. We will show that $\Gamma(T_2) > \Gamma(T_3)$ in Lemma C.2.

     (c) $T_3$ to $T_4$. The sub-trees of $T_3$ become BBTs in $T_4$. For the same reason as (a), we have $\Gamma(T_3) \geq \Gamma(T_4)$.

Putting them together, we get $\Gamma(T_1) > \Gamma(T_4)$ and Lemma C.1 follows.

□

**Lemma C.2.** *After swapping $V_{lr}$ and $V_{rl}$, we obtain $T_3$ from $T_2$, for which $\Gamma(T_2) > \Gamma(T_3)$.*

*Proof.* We only need to consider the changes in cost of three nodes: root and its left and right children, since the cost contributed by each of the remaining nodes does not change after swapping. Ignoring the unchanged costs, define

$$\text{cost}(T_2) \;=\; n_l n_r \log n + n_{ll} n_{lr} \log n_l + n_{rl} n_{rr} \log n_r$$

$$=\; n_l n_r \log n + \left\lfloor \frac{n_l}{2} \right\rfloor \left\lceil \frac{n_l}{2} \right\rceil \log n_l + \left\lceil \frac{n_r}{2} \right\rceil \left\lfloor \frac{n_r}{2} \right\rfloor \log n_r,$$

where $n_l = n_{ll} + n_{lr}, n_r = n_{rl} + n_{rr}$. Both of them are at least 1. Similarly, define

$$\text{cost}(T_3) \;=\; (n_{ll} + n_{rl})(n_{lr} + n_{rr}) \log n + n_{ll} n_{rl} \log (n_{ll} + n_{rl}) + n_{lr} n_{rr} \log (n_{lr} + n_{rr})$$

$$=\; \left\lfloor \frac{n}{2} \right\rfloor \left\lceil \frac{n}{2} \right\rceil \log n + \left\lfloor \frac{n_l}{2} \right\rfloor \left\lceil \frac{n_r}{2} \right\rceil \log \left( \left\lfloor \frac{n_l}{2} \right\rfloor + \left\lceil \frac{n_r}{2} \right\rceil \right) + \left\lceil \frac{n_l}{2} \right\rceil \left\lfloor \frac{n_r}{2} \right\rfloor \log \left( \left\lceil \frac{n_l}{2} \right\rceil + \left\lfloor \frac{n_r}{2} \right\rfloor \right)$$

556 Denote

$$
\begin{aligned}
\Delta &= \Gamma(T_2) - \Gamma(T_3) \\
&= \text{cost}(T_2) - \text{cost}(T_3) \\
&= \left\lfloor \frac{n_l}{2} \right\rfloor \left\lceil \frac{n_l}{2} \right\rceil \log\left(\frac{n_l}{n}\right) + \left\lceil \frac{n_r}{2} \right\rceil \left\lfloor \frac{n_r}{2} \right\rfloor \log\left(\frac{n_r}{n}\right) \\
&\quad - \left\lfloor \frac{n_l}{2} \right\rfloor \left\lceil \frac{n_r}{2} \right\rceil \log\left(\frac{\left\lfloor \frac{n_l}{2} \right\rfloor + \left\lceil \frac{n_r}{2} \right\rceil}{n}\right) - \left\lceil \frac{n_l}{2} \right\rceil \left\lfloor \frac{n_r}{2} \right\rfloor \log\left(\frac{\left\lceil \frac{n_l}{2} \right\rceil + \left\lfloor \frac{n_r}{2} \right\rfloor}{n}\right) \quad (2)
\end{aligned}
$$

557 So we only have to show that $\Delta > 0$. We consider the following three cases according to the odevity
558 of $n_l$ and $n_r$.

559 **Case** 1: $n_l$ and $n_r$ are even.

560 **Case** 2: $n_l$ and $n_r$ are odd.

561 **Case** 3: $n_l$ is odd while $n_r$ is even.

562 The case that $n_l$ is even while $n_r$ is odd is symmetric to **Case** 3.

563 For **Case** 1, if both $n_l$ and $n_r$ are even, then notations of rounding in Eq. (2) can be removed and $\Delta$
564 can be simplified as

$$
\Delta = \frac{n_l^2}{4} \log\left(\frac{n_l}{n}\right) + \frac{n_r^2}{4} \log\left(\frac{n_r}{n}\right) + \frac{n_l n_r}{2}.
$$

565 Let $p = n_l/n, q = n_r/n$, and so $p + q = 1$. Recall that $T_1$ is unbalanced on the root $\lambda$, so is $T_2$.
566 Thus $p \neq q$. Multiplying by $\frac{4}{n^2}$ on both sides, we only have to prove that

$$
p^2 \log p + q^2 \log q + 2pq > 0.
$$

That is,

$$
\frac{p}{q} \log p + \frac{q}{p} \log q + 2 > 0.
$$

567 Let $g(x) = \frac{x}{1-x} \log x$. Then we only need to show that $g(p) + g(q) + 2 > 0$ when $p \neq q$. Since

$$
\begin{aligned}
g'(x) &= \frac{(1-x) + \ln x}{\ln 2 \cdot (1-x)^2}, \\
g''(x) &= -\frac{x^2 - 2x \ln x - 1}{\ln 2 \cdot x(1-x)^3}.
\end{aligned}
$$

It is easy to check that $g''(x) > 0$ when $0 < x < 1$. So $g(x)$ is strictly convex in the interval $(0,1)$.
Since $p \neq q$,

$$
g(p) + g(q) > 2g\left(\frac{p+q}{2}\right) = -2.
$$

568 Thus $\Delta > 0$ holds.

569 For **Case** 2, if both $n_l$ and $n_r$ are odd, then $\Delta$ can be split into two parts $\Delta = \Delta_1 + \Delta_2$, in which

$$
\begin{aligned}
\Delta_1 &= \frac{n_l^2}{4} \log\left(\frac{n_l}{n}\right) + \frac{n_r^2}{4} \log\left(\frac{n_r}{n}\right) + \frac{n_l n_r}{2} \\
\Delta_2 &= -\frac{1}{4} \log\left(\frac{n_l}{n}\right) - \frac{1}{4} \log\left(\frac{n_r}{n}\right) - \frac{1}{2}
\end{aligned}
$$

570 Since we have shown that $\Delta_1 > 0$, if we can prove $\Delta_2 \geq 0$, then the lemma will hold for **Case** 2.
571 Due to the convexity of logarithmic function, this holds clearly since

$$
2 \log\left(\frac{n}{2}\right) \geq \log n_l + \log n_r.
$$

572 For **Case** 3, if $n_l$ is odd while $n_r$ is even,

$$
\Delta = \frac{n_l^2 - 1}{4} \log\left(\frac{n_l}{n}\right) + \frac{n_r^2}{4} \log\left(\frac{n_r}{n}\right) - \left[\frac{(n_l - 1)n_r}{4} \log\left(\frac{n-1}{2n}\right) + \frac{(n_l + 1)n_r}{4} \log\left(\frac{n+1}{2n}\right)\right].
$$

573 Multiplying the above equation by $4\ln 2$, without changing its sign, yields

$$(4\ln 2)\Delta = (n_l^2 - 1)\ln\left(\frac{n_l}{n}\right) + n_r^2 \ln\left(\frac{n_r}{n}\right) - \left[(n_l - 1)n_r \ln\left(\frac{n-1}{2n}\right) + (n_l + 1)n_r \ln\left(\frac{n+1}{2n}\right)\right]$$

574 Splitting the right hand side into two parts,

$$
\begin{aligned}
A &= n_l^2 \ln\left(\frac{n_l}{n}\right) + n_r^2 \ln\left(\frac{n_r}{n}\right) + 2n_l n_r \ln 2 \\
B &= -\ln\left(\frac{n_l}{n}\right) - (n_l + 1)n_r \ln\left(1 + \frac{1}{n}\right) - (n_l - 1)n_r \ln\left(1 - \frac{1}{n}\right)
\end{aligned}
$$

575 Since $n$ is odd and the root $\lambda$ of $T_2$ is unbalanced, we only need to consider the case that $n_l = $
576 $(n-i)/2$, $n_r = (n+i)/2$ (Note that $n_l$ and $n_r$ are symmetric. So if $(n-i)/2$ is even, exchange
577 $n_l$ and $n_r$), where both $n$ and $i$ are odd satisfying $n > i \geq 3$. Next we show that in this case,
578 $A \geq \ln(1/5) + 4^2 \ln(4/5) + 2 \cdot 4\ln 2$ and $B > \ln 2 - 3/4 - (2/3) \cdot (1/5^2)$. By calculation,
579 $\Delta = A + B > 0$ for **Case** 3.

580 **Claim C.1.** $A \geq \ln(1/5) + 4^2 \ln(4/5) + 2 \cdot 4\ln 2$ *for odd integers* $n > i \geq 3$.

581 *Proof.* Substituting $n_l = (n-i)/2$, $n_r = (n+i)/2$ into the $A$ yields

$$A = C(n, i) \triangleq \left(\frac{n-i}{2}\right)^2 \ln\left(\frac{n-i}{2n}\right) + \left(\frac{n+i}{2}\right)^2 \ln\left(\frac{n+i}{2n}\right) + 2 \cdot \frac{n-i}{2} \cdot \frac{n+i}{2} \ln 2.$$

582 Treat $n$ as a continuous variable, we have

$$\frac{\partial C(n, i)}{\partial n} = \frac{1}{2}\left[(n+i)\ln\left(1 + \frac{i}{n}\right) + (n-i)\ln\left(1 - \frac{i}{n}\right) - \frac{i^2}{n}\right]$$

583 Multiplying the above equation by $2/n$ and setting $x = i/n$ yields

$$
\begin{aligned}
f(x) &\triangleq (1+x)\ln(1+x) + (1-x)\ln(1-x) - x^2, \\
f'(x) &= \ln(1+x) - \ln(1-x) - 2x, \\
f''(x) &= \frac{2x^2}{1 - x^2}.
\end{aligned}
$$

584 It is easy to check that $f(0) = 0$ and $f'(0) = 0$. When $0 < x < 1$, $f''(x) > 0$. Thus $f'(x) > 0$ and
585 $f(x) > 0$. This means that $\partial C(n, i)/\partial n > 0$ for all $n > 0$. So $C(n, i) \geq C(i + 2, i)$ for $n \geq i + 2$
586 (When $i$ is fixed, the minimum value of $n$ can be taken to $i + 2$, which makes $n_l = (n-i)/2$ and
587 $n_r = (n+i)/2$ integral). The curves of $C(n, i)$ for varying $i$ are plotted in Figure 2.

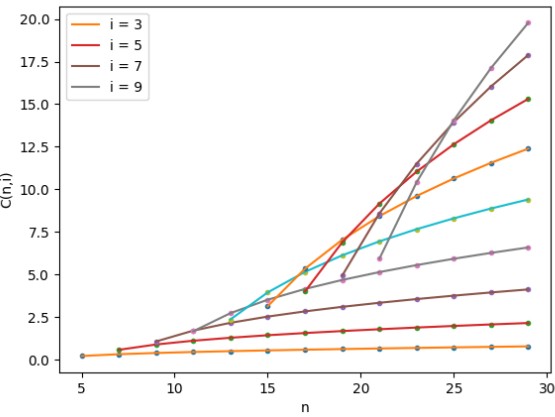

Figure 2: Functions $C(n, i)$

When $n = i + 2$, we get $n_l = (n - i)/2 = 1$ and $n_r = (n + i)/2 = n - 1$. Substituting them into $A$ yields

$$D(n) \triangleq \ln\left(\frac{1}{n}\right) + (n-1)^2 \ln\left(1 - \frac{1}{n}\right) + 2(n-1)\ln 2,$$

$$\frac{dD}{dn} = 1 - \frac{2}{n} + 2\ln 2 + 2(n-1)\ln\left(1 - \frac{1}{n}\right).$$

When $n > 2$, it is easy to check that $dD/dn > 0$. So the minimum value of $d(n)$, which is also the minimum value of $C(i+2, i)$, is achieved at $n = i + 2 = 5$. So $A = C(n, i) \geq C(i+2, i) \geq C(5, 3) = \ln(1/5) + 4^2 \ln(4/5) + 2 \cdot 4 \ln 2$. $\qquad \square$

**Claim C.2.** $B > \ln 2 - 3/4 - (2/3) \cdot (1/5^2)$.

*Proof.* Due to the facts that

$$\ln\left(1 + \frac{1}{n}\right) < \frac{1}{n} - \frac{1}{2n^2} + \frac{1}{3n^3},$$

$$\ln\left(1 - \frac{1}{n}\right) < -\frac{1}{n} - \frac{1}{2n^2} - \frac{1}{3n^3},$$

we have

$$\begin{aligned} B &= -\ln\left(\frac{n_l}{n}\right) - (n_l + 1)n_r \ln\left(1 + \frac{1}{n}\right) - (n_l - 1)n_r \ln\left(1 - \frac{1}{n}\right) \\ &> -\ln\left(\frac{n_l}{n}\right) + \frac{n_l n_r}{n^2} - \frac{2n_r}{n} - \frac{2n_r}{3n^3} \\ &> -\ln\left(\frac{n_l}{n}\right) + \frac{n_l n_r}{n^2} - \frac{2n_r}{n} - \frac{2}{3n^2}. \end{aligned}$$

Let $\alpha = n_l/n$, then

$$\begin{aligned} B &> -\ln \alpha + \alpha(1 - \alpha) - 2(1 - \alpha) - \frac{2}{3n^2} \\ &\geq \ln 2 - \frac{3}{4} - \frac{2}{3n^2}. \end{aligned}$$

When $n \geq 5$, $B > \ln 2 - 3/4 - (2/3) \cdot (1/5^2)$. $\qquad \square$

Combining Claims C.1 and C.2, Lemma C.2 follows. $\qquad \square$

This completes the proof of Proposition 2.2.

# D    Proof of Theorem 3.1

*Proof.* Note that $\text{cost}^T(G)$ for any cluster tree $T$ has a trivial upper bound. That is,

$$\text{cost}^T(G) = \sum_{e \in E} \text{cost}^T(e) \leq \sum_{e \in E} w_e \cdot \log(\text{vol}(G)) \leq \frac{\text{vol}(G) \cdot \log(\text{vol}(G))}{2},$$

where $\text{cost}^T(e) = w_e \log \text{vol}(\text{LCA}_T(e))$. Let $T^*$ be the optimal cluster tree that achieves the minimum cost, we present here a lower bound for $\text{cost}^{T^*}(G)$. Referring to the dense branch technique [10, 15], we start with the root node $A_0$ and walk along $T^*$ recursively as follows: at every internal node $A_i$, walk down to the node $A_{i+1}$ of higher volume between its two children. This process stops when we reach node $A_k$ such that $\text{vol}_G(A_k) \leq \frac{2\text{vol}(G)}{3}$. Denote $A \triangleq A_k$ as well as $B \triangleq V \backslash A_k$. By construction, it holds that $\text{vol}_G(A) > \frac{\text{vol}(G)}{3}$ and $\text{vol}_G(B) \geq \frac{\text{vol}(G)}{3}$. Moreover, $\text{vol}_G(A_i) > \frac{2\text{vol}(G)}{3}$

for every $0 \le i < k$. The basic idea behind the dense branch is that the $cut(A,B)$ has a significant contribution to $cost^{T^*}(G)$.

$$cost^{T^*}(G) = \sum_{e=\{u,v\}} w_e \cdot \log(\text{vol}_G(u \vee v))$$

$$\ge \sum_{\substack{e=\{u,v\} \\ e \in E\{A,B\}}} w_e \cdot \log(\text{vol}_G(u \vee v))$$

$$\ge w(A,B) \cdot \log\left(\frac{2\text{vol}(G)}{3}\right).$$

$$\ge \Phi(G) \cdot \frac{\text{vol}(G)}{3} \cdot \log\left(\frac{2\text{vol}(G)}{3}\right).$$

Let $T$ be an arbitrary cluster tree, and $T^*$ be an optimal tree. We have

$$\frac{cost^T(G)}{cost^{T^*}(G)} \le \frac{3}{2\Phi(G)} \cdot \frac{\log(\text{vol}(G))}{\log\left(\frac{2\text{vol}(G)}{3}\right)} = O(\Phi(G)^{-1}).$$

□

# E    Proof of Theorem 3.2

*Proof.* To prove Theorem 3.2, we only have to prove the following lemma. Then the theorem follows from a simplification of the approximation factor.

**Lemma E.1.** *Let* $\alpha = \max_i\{\Phi_G(P_i)\}$ *and* $\beta = \min_i\{\Phi(G[P_i])\}$. *Algorithm 2 achieves* $\left(\left(\left(\log\left(\frac{1}{1-\alpha}\right) + 1\right) + \frac{2\alpha}{1-\alpha}\left(1 + \log\frac{k}{1-\alpha}\right)\right) \cdot \frac{3}{2\beta \log\left(\frac{4}{3}\right)}\right)$-*approximation.*

*Proof.* We group the edges of G into two categories: let $E_1$ be the set of edges in the induced subgraphs $G[P_i]$ for all $1 \le i \le l$, i.e.,

$$E_1 \triangleq \cup_{i=1}^l E[G[P_i]],$$

and $E_2$ be the remaining crossing edges. Then we have

$$cost^T(G) = \sum_{e \in E_1} cost^T(e) + \sum_{e \in E_2} cost^T(e).$$

We denote by $\text{vol}(G[P_i])$ the volume of the induced graph $G[P_i]$, by $\text{vol}_G(P_i)$ the volume of $P_i$ in $G$, and by $\text{parent}^T(P_i)$ the parent of $P_i$ on $T$. Then it holds for every $P_i$ that

$$\text{vol}_G(\text{parent}^T(P_i)) \le k \cdot \text{vol}_G(P_i).$$

By the construction of $T$ we have that

$$\text{vol}_G(\text{parent}^T(P_i)) = \sum_{j=1}^i \text{vol}_G(P_j) \le i \cdot \text{vol}_G(P_i) \le k \cdot \text{vol}_G(P_i).$$

Note that

$$w(P_i, V\backslash P_i) = \text{vol}_G(P_i) - \text{vol}(G[P_i]) \le \alpha \cdot \text{vol}_G(P_i),$$

$$(1-\alpha) \cdot \text{vol}_G(P_i) \le \text{vol}(G[P_i]),$$

and thus

$$\text{vol}_G(\text{parent}^T(P_i)) \le k \cdot \text{vol}_G(P_i) \le \frac{k}{1-\alpha}\text{vol}(G[P_i]).$$

Combining the above, we have that

$$
\begin{aligned}
\sum_{e \in E_1} \mathrm{cost}^T(e) \ &\leq \ \sum_{e \in E_1} w_e \cdot \log(\mathrm{vol}_G(P_i)) \\
&\leq \ \sum_{e \in E_1} w_e \cdot \log\left( \frac{1}{1-\alpha} \mathrm{vol}(G[P_i]) \right) \\
&= \ \sum_{e \in E_1} \left( w_e \cdot \log \frac{1}{1-\alpha} + w_e \cdot \log(\mathrm{vol}(G[P_i])) \right) \\
&\leq \ \left( \log \frac{1}{1-\alpha} + 1 \right) \cdot \sum_{j=1}^{k} \frac{\mathrm{vol}(G[P_i]) \cdot \log(\mathrm{vol}(G[P_i]))}{2},
\end{aligned}
$$

and

$$
\begin{aligned}
\sum_{e \in E_2} \mathrm{cost}^T(e) \ &\leq \ \sum_{j=1}^{k} w(P_i, V \backslash P_i) \cdot \log(\mathrm{vol}_G(\mathrm{parent}^T(P_i))) \\
&\leq \ \sum_{j=1}^{k} \frac{\alpha}{1-\alpha} \mathrm{vol}(G[P_i]) \log\left( \frac{k}{1-\alpha} \mathrm{vol}(G[P_i]) \right) \\
&\leq \ \sum_{j=1}^{k} \frac{\alpha}{1-\alpha} \left( 1 + \log \frac{k}{1-\alpha} \right) \mathrm{vol}(G[P_i]) \log(\mathrm{vol}(G[P_i])) \\
&= \ \frac{2\alpha}{1-\alpha} \left( 1 + \log \frac{k}{1-\alpha} \right) \cdot \sum_{j=1}^{k} \frac{\mathrm{vol}(G[P_i]) \cdot \log(\mathrm{vol}(G[P_i]))}{2}.
\end{aligned}
$$

Let $T^*$ be the optimal cluster tree of $G$, and $OPT_G$ be the optimal value. We have

$$
OPT_G = \mathrm{cost}_G(T^*) \geq \sum_{i=1}^{l} \sum_{e \in E(G[P_i])} \mathrm{cost}_{T^*}(e) \geq \sum_{i=1}^{l} OPT_{G[P_i]}.
$$

Denote by

$$
h(\alpha, k) = \left( \left( \log\left( \frac{1}{1-\alpha} \right) + 1 \right) + \frac{2\alpha}{1-\alpha} \left( 1 + \log \frac{k}{1-\alpha} \right) \right).
$$

We have

$$
\begin{aligned}
\mathrm{cost}^T(G) \ &= \ \sum_{e \in E_1} \mathrm{cost}^T(e) + \sum_{e \in E_2} \mathrm{cost}^T(e) \\
&\leq \ h(\alpha, k) \cdot \sum_{j=1}^{k} \frac{\mathrm{vol}(G[P_i]) \cdot \log(\mathrm{vol}(G[P_i]))}{2} \\
&\leq \ h(\alpha, k) \cdot \sum_{j=1}^{k} \frac{\mathrm{vol}(G[P_i]) \cdot \log(\mathrm{vol}(G[P_i]))}{2\Phi_{G[P_i]} \cdot \frac{1}{3}\mathrm{vol}(G[P_i]) \cdot \log(\frac{2}{3}\mathrm{vol}(G[P_i]))} OPT_{G[P_i]} \\
&\leq \ h(\alpha, k) \cdot \max_i \frac{3 \log(\mathrm{vol}(G[P_i]))}{2\Phi_{G[P_i]} \cdot \log(\frac{2}{3}\mathrm{vol}(G[P_i]))} \sum_{j=1}^{k} OPT_{G[P_i]} \\
&\leq \ h(\alpha, k) \cdot \max_i \frac{3 \log(\mathrm{vol}(G[P_i]))}{2\Phi_{G[P_i]} \cdot \log(\frac{2}{3}\mathrm{vol}(G[P_i]))} OPT_G \\
&\leq \ h(\alpha, k) \cdot \frac{3}{2\beta \log(\frac{4}{3})} OPT_G
\end{aligned}
$$

Lemma E.1 follows. □

Note that $h(\alpha, k) = O\left( \frac{1}{(1-\alpha)} \log \frac{k}{1-\alpha} \right)$, Theorem 3.2 follows. □

# F Experimental results on Amazon network

We do our experiments on Amazon network [4] for which the set of ground-truth clusters has been given. For two sets $A, B$, the *Jaccard Index* of them is defined as $J(A, B) = |A \cap B|/|A \cup B|$. We pick the largest cluster which is a subgraph with $58283$ vertices and $133178$ edges. We run HCSE algorithm on it. For each ground-truth cluster $c$ that appears in this subgraph, we find from the resulting cluster tree an internal node that has maximum Jaccard index with $c$. Then we calculate the average Jaccard index $\overline{J}$ over all such $c$. We also calculate cost(SE) and cost(Das). The results are demonstrated in Table 3. HCSE performs better for $\overline{J}$ and cost(SE), while LOUVAIN performs better for cost(Das). Because of unbalance in over-fitting and under-fitting traps, HLP outperforms none of the other two algorithms for all criteria.

| index | HCSE | HLP | LOUVAIN |
|---|---|---|---|
| $\overline{J}$ | **0.20** | 0.16 | 0.17 |
| cost(SE) | **1.85E6** | 2.05E6 | 1.89E6 |
| cost(Das) | 5.57E8 | 3.99E8 | **3.08E8** |

Table 3: Comparisons of the average Jaccard index ($\overline{J}$), cost function based on structural entropy (cost(SE)) and Dasgupta's cost function (cost(Das)).

# G Some figures and pseudocodes

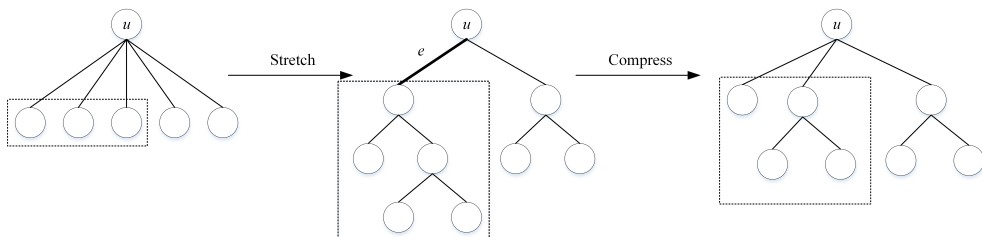

Figure 3: Illustrations of stretch and compress for a $u$-triangle. A binary cluster tree is constructed first by stretch, and then edge $e$ is compressed, which yields a non-binary tree.

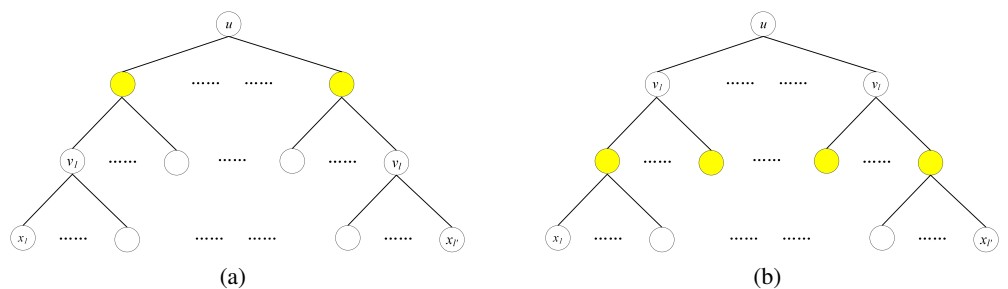

Figure 4: Illustration of stratification for a 2-level cluster tree. The preference of (a) and (b) depends on the average sparsity of triangles at each level.

---

[4]http://snap.stanford.edu/data/

**Algorithm 5:** Stretch

---

**Input:** a $u$-triangle $T_u$
**Output:** a binary tree rooted at $u$

1 Let $\{v_1, v_2, \ldots, v_\ell\}$ be the set of leaves of $T_u$;
2 Compute $\eta(a, b)$ which is the cost reduced by merging siblings $a, b$ into a single cluster;
3 **for** $t \in [\ell - 1]$ **do**
4    $(\alpha, \beta) \leftarrow \arg\max_{(a,b) \text{ are siblings}}\{\eta(a, b)\}$;
5    Add a new node $\gamma$;
6    $\gamma.parent \leftarrow \alpha.parent$;
7    $\alpha.parent = \gamma$;
8    $\beta.parent = \gamma$;

9 return $T_u$

---

**Algorithm 6:** Compress

---

**Input:** a binary tree $T$

1 **while** *T's height is more than* 2 **do**
2    $e \leftarrow \arg\min_{e' \in \hat{E}(T)}\{\Delta(e')\}$;
3    Denote $e = (u, v)$ where $u$ is the parent of $v$;
4    **for** $w \in v.children$ **do**
5       $w.parent \leftarrow u$;
6    Delete $v$ from $T$;

---

