# OpenReview forum: "An Information-theoretic Perspective of Hierarchical Clustering"
_NeurIPS.cc/2022/Conference — NeurIPS 2022 Submitted_

### Official Review · Reviewer_uGUC · 2022-07-04

**Rating:** 4
**Confidence:** 4
**Soundness:** 2 fair
**Presentation:** 2 fair
**Contribution:** 3 good

**Summary:**

This paper proposes to use the structural entropy as a cost function in hierarchical clustering. Theoretical analysis shows that the proposed cost function implicitly prefers balanced clusters, which is different from the existing cost functions and a remarkable feature of the proposal. Moreover, an approximation algorithm with its theoretical guarantee is presented. Furthermore, a more realistic efficient algorithm is also proposed and it is empirically examined on synthetic and real-world datasets.

**Questions:**

It would be great if you could address my concerns listed in the quality section (motivation, assumption) and the significance section (empirical evaluation), and clarify each of the points in the clarity section.


**Limitations:**

Potential negative societal impact is not described. it would be better to discuss it with respect to forcing balanced hierarchical clusters.


**Strengths And Weaknesses:**

## Originality

The structural entropy is originally defined on a graph with a partition, hence its application to hierarchical clustering, proposed in this paper, seems to be straightforward.
Although its theoretical analysis, including Proposition 2.2 and Theorem 3.2, is interesting, the overall originality of this paper is not so remarkable since it is a straightforward combination of [8], [10], and [14].
Of course, such a combination definitely has a potential of a significant contribution, while currently it is not well demonstrated as stated in the following sections.

## Quality

The motivation is not so clear. I understand that the proposal is suitable for finding balanced clusters, while its importance in the context of hierarchical clustering is not well discussed. That is, why do we need to find balanced clusters? What kind of application do we have for balanced hierarchical clustering? I would recommend carefully discussing the motivation of the proposal in this paper.

In addition, the effectiveness of balanced hierarchical clustering is not empirically examined in experiments. Please present a scenario with synthetic and/or real-world datasets in which balanced clusters are preferable and the proposal actually can find such clusters with better performance compared with existing methods.

In the problem setting, any undirected weighted graph is in the scope of this paper. However, in theoretical analysis, it is always assumed that a given graph is a cardinality weighted graph. Does this mean that the presented theoretical results only apply to a special case of graphs? How restrictive is this assumption?

## Clarity

There are a number of unclear points:

- The definition of $g_\alpha$ is not clear. In line 137, it is explained as "$g_\alpha$ denotes the sum of weights of edges in $G$ with exactly one end-point in the set of vertices corresponding to $\alpha$". Does this mean $g_\alpha$ is equivalent to $d_\alpha$? Also, it is used as $(u, v) \in g_\alpha$ in the proof of Proposition 2.1. It is confusing as $g_\alpha$ should be a real number. Please revise it.

- In line 143, it is said that $g(x, y) = x + y$, which means that $g(|u|, |v|) = |u| + |v|$. Although it is said that $g(|u|, |v|)$ is a generalized term of $|u \vee v|$, I think in general $|u| + |v| = |u \vee v|$ does not hold. What is the reasoning of this?

- In the paragraph immediately preceding Prop.2.2, the authors discuss the difference between the proposed cost function and that in [8]. Does this discussion mean that the proposal does not satisfy the desirable condition "the cost for every binary cluster tree of an unweighted clique is identical"? If so, a more careful discussion of why the balanced clusters is more important than the above condition in practical situations would be desirable.

- In line 182, I think $\log(u \vee v)$ should be $\log vol(u \vee v)$.

- In Theorem 3.1, the conductance $\Phi_G$ is used without any definition. Please define it beforehand.

- In Theorem 3.2, what is the definition of $O(\cdot)$-approximation in terms of hierarchical clustering?

- In experiments, it is not clear to me how to generate ground-truth clusters. Why is the length of $\vec{p}$ fixed to 4? Does this mean that the level of hierarchy is always fixed to 4 regardless of the sample size?

## Significance

From the theoretical viewpoint, this paper theoretically analyzes the property of the proposed cost function in Prop. 2.1 and 2.2., and carefully design its approximation algorithm with its theoretical guarantee in Theorem 3.2. I think this is a good contribution and can be valuable in the task of hierarchical clustering.

From the empirical viewpoint, the significance is not high as the proposed algorithm is not directly examined.
In experiments, Algorithms 3 and 4 are used as the proposed method, while they are different from Algorithms 1 and 2, which are the key contribution of this paper and carefully theoretically analyzed.
Therefore, to empirically examine the proposal, Algorithms 1 and 2 should be used in experiments. If they cannot be used for some reason, at least the relationship between Algorithms 1&2 and 3&4 should be presented.

In addition to the above issue, in experiments, using real-world datasets with the ground-truth labels is strongly recommended for evaluation.
I understand that it is difficult to obtain ground-truth hierarchies, while it is possible to collect datasets with ground-truth flat partitions (that is, labels). Then, for example, one can use the dendrogram purity to quantify the goodness of obtained hierarchies. See: Kobren et al.,  A hierarchical algorithm for extreme clustering, SIGKDD2017.

---

> ### Author Response · Authors · 2022-08-02
> **Response to the concerns and questions.**
>
> Thank the reviewer for the insightful comments. For the weakness and questions, our responses follow.
>
> For the motivation of our study, our major purpose is to provide an information-theoretic perspective for the hierarchical clustering and bridge it with the widely used combinatorial definitions based on Dasgupta’s cost function. One advantage of our cost function compared with all existing objectives is that our cost consider balance as a factor naturally. Our cost is essentially a trade-off between the two factors: connection and balance. In Prop. 2.2, we showed that when connections become uniform (in cliques), the factor of balance determines the optima of cluster trees. Balance of cluster trees is an important factor in clustering study. It has many applications in practice, for example, load balancing for cloud computing and management, network design, and avoidance of outlier cluster formation, etc. Balance has been widely considered in partitioning and clustering problems, which has motivated some classic optimization problems such as balanced cut and balanced clustering (e.g., balanced k-means), etc. We thought it is a common sense that balancing is a fundamental issue in clustering problem and thus didn’t spend much space for the discussion of its motivation.
>
> For the experimental evaluation for the effectiveness of balanced hierarchical clustering, the ground truth clusters in HSBM are of equal size, and the high NMI values in Table 1 demonstrate that the clusters in most levels on the balanced ground truth cluster trees have been identified. A reliable experiment to show the difference of cost(SE) and cost(Das) in the balance factor should be designed carefully. Perhaps an experiment on HSBM with various sizes of clusters will be helpful.
>
> For the cardinality setting of weighted graphs, note that the problem of approximating structural entropy defined in Line 136 is well defined since structural entropy is always non-negative. However, the equivalent optimization problem to minimize cost(SE) defined in Eq. (1) may have negative values since it is related to the scale of edge weights. This makes the approximation problem not well-defined. When the edge weights are at least 1, cost(SE) is always non-negative, and we study approximation algorithms in this well-defined case only. However, the proposed algorithms that treat cost(SE) as the objective indeed work for any cases. The restriction of the assumption is that the approximation factors shown in Theorems 3.1 and 3.2 can be guaranteed in the well-defined settings.
>
> For the relationship of Algorithms 1&2 with Algorithms 3&4, The former two are binary clustering methods while the latter two belong to a framework for non-binary clustering. Any binary clustering algorithm can be incorporated with this framework by serving for the stretch step. In HCSE, we didn’t use Algorithm 1&2 because the spectral partitioning algorithm that is recalled as a subroutine is quite time-consuming. According to the intuition behind our framework (stated in Section 4, Lines 236-241), any (rough) binary clustering algorithm that can see the sparsest level after stretch will be reasonable to be incorporated in our framework. Thus, for computational efficiency, we adopt a simple HAC binary clustering method for experiments.
>
> For the evaluation of experiments on real datasets, it is indeed worth having more consideration in experiment design. Perhaps, the purity of dendrogram is a good criterion for the evaluation of cluster trees when only the ground truth of flat or incomplete clusters is given.
>
> For the questions on clarity, we answer them as follows:
>
> (1) For an internal node $\alpha$ on a cluster tree, we did not define $d_\alpha$. We keep the definition of $g_\alpha$ and have revised the statement in the proof of Prop. 2.1.
>
> (2) We have corrected the statements in Line 143: “[8] generalizes the term $|u\vee v|$ in the definition of $c^T(G)$ to be a general function $g(|L|,|R|)$, where $L$ and $R$ are the two children of $u\vee v$, respectively.”
>
> (3) Our cost(SE) indeed violates the condition "the cost for every binary cluster tree of an unweighted clique is identical" that is proposed in [8]. As we stated above, we think that connection and balance are two factors we need to consider in clustering problem. This condition only consider connection while balance has been totally ignored. In contrast, cost(SE) is a trade-off between these two factors.
>
> (4) Yes, it is a typo. We have corrected it.
>
> (5) $\Phi(G)$ is in fact $\Phi_G$ in Theorem 3.1. We have unified them.
>
> (6) We have clarified in Theorem 3.2 that it is an approximation factor for $\text{cost}^T(G)$.
>
> (7) In this experiment, we choose the length of vector $\vec{p}$ as 4 as an example, which means that the number of levels in HSBM is 4. It is indeed better to have more choices for the number of hierarchies and graph size.

---

> > ### Comment · Reviewer_uGUC · 2022-08-07
> > **Thank you for the response**
> >
> > I thank the authors for their response.
> >
> > Although my concerns are partially resolved and I think there exists an interesting contribution in this paper, I still feel that the paper is not ready for publication due to the lack of significance (my two concerns remain). In addition, some of the concerns in the clarity section are still unclear.
> >
> > Therefore I would like to keep my score.

---

### Official Review · Reviewer_xKCS · 2022-07-09

**Rating:** 4
**Confidence:** 3
**Soundness:** 2 fair
**Presentation:** 2 fair
**Contribution:** 3 good

**Summary:**

The manuscript proposes an information-theoretic cost function for hierarchical clustering that is based on structural entropy. Based on the theoretical analysis of the cost function, the authors propose algorithms for special classes of graphs (Algorithm 1 for expander-like graphs and Algorithm 2 for well-clustered graphs) as well as for general graphs (Algorithms 3 and 4). In their experiments they show that their method outperforms modularity- and label propagation-based clustering in terms of finding the correct number of hierarchies.

**Questions:**

1) In line 143, the usage of $g(|u|,|v|)$ is not clear. If $u$ and $v$ are vertices, as suggested by the sum in line 141, then what is the meaning of $|u|$?
2) What is an ultrametric (line 170)?
3) Should line 10 in Algorithm 3 really be indented?
4) (How) Is $\Delta_t\mathcal{H}$ in lines 282-286 connected with $\Delta\mathcal{H}(u)$?
5) Referring to Fig. 1, the text claims that the inflection points are at $t=4$ -- I do see two more inflection points at $t=2$ for two of the datasets. How should they be accounted for?

**Limitations:**

While the authors did not discuss the limitations of their work, they pointed out interested directions for future research. An important limitation in my opinion is clearly the lack of extensive experimental evidence (as pointed out above).

**Strengths And Weaknesses:**

The paper treats an interesting topic and proposes, to the best of my knowledge, a novel cost function. The algorithmic approach (stretch and compress) seems very promising. The paper is theoretically deep and accompanies cost function and algorithm with performance bounds and runtime analyses.

What prevents me from assigning a better score is the fact that the experiments are not convincing. While the results in Table 1 are promising, I am confused by the results in Table 2. There it is shown that LOUVAIN performs better in terms of the cost function on which HCSE is based, occasionally also for the same number of hierarchies. This could indicate that not the cost function yields improved performance, but rather its algorithmic implementation. A simple check would be to accompany Table 1 with cost(SE) values, but I also suggest more in-depth ablation analyses (e.g., replace the optimization of cost(SE) by a different algorithm, potentially a global optimization heuristic). Further, experiments on hierarchical graphs with ground truth would be welcome.

Second, the authors claim that their approach is interpretable, or even more interpretable than hierarchical Louvain or label propagation (lines 80-81, 89-90, and Section 6). I do not fully agree. On the one hand, the proposed cost function based on structural entropy is not easily accessible, at least not when compared to modularity. And second, the algorithmic implementation (stretch and compress) is, in my opinion, less interpretable than the procedure of propagating labels. In my opinion, the claim of interpretability seems not justified.

A final shortcoming of the manuscript is that it is not always clear. For example, the title and abstract are not clear about whether we consider hierarchical clustering for graphs or for general data (but see also questions below). Several paragraphs are hard to parse, e.g., lines 33-36, 51-58, 229-232, 239-241, and 268-270, and terms like "pathosis" (line 183) or phrases like "when $\phi(G)$ is as large as a constant" are not perfectly clear. This is not a critical, though, as I believe that these shortcomings can be repaired easily. While revising, I further suggest proof reading by an English native speaker.

---

> ### Author Response · Authors · 2022-08-02
> **Response to the concerns and questions.**
>
> Thank the reviewer for the insightful comments. For the weakness and questions, our responses follow.
>
> For the significance of our second experiments, we need to verify the performance of our framework for non-binary clustering that can be incorporated with any binary clustering algorithm in the stretch step. The experiments on real datasets demonstrate that our explainable framework can generate quality cluster trees that are competitive with what the popular LOUVAIN and HLP output. The number of levels obtained by HCSE is more persuasive than those found by LOUVAIN and HLP which is solely based on a bottom-up framework, since HCSE follows an interpretable framework in finding the natural number of levels.
>
> For the ablation analysis in different cost functions, according to the intuition behind our framework (stated in Section 4, Lines 236-241), any (rough) binary clustering algorithm that can see the sparsest level after stretch will be reasonable to be incorporated in our framework. We do not focus on the performance of various cost functions, but need to verify the new framework. In fact, the cost values of both cost(SE) and cost(Das), or any other existing costs, cannot evaluate a non-binary cluster tree since a blind pursuit of such kind of cost will yield a binary tree, as we discussed in the future discussions. The values listed in the experiments on real datasets demonstrate that our algorithm HCSE has a competitive effectiveness with LOUVAIN and HLP. We also think that the number of levels obtained by HCSE is more persuasive than those found by LOUVAIN and HLP (although hard to verify), since HCSE follows an interpretable framework in finding the natural number of levels. However, it is better to do more experiments on hierarchical graphs with ground truth.
>
> For the interpretability of our framework, we don’t think a bottom-up non-binary clustering framework is interpretable, because there is no criterion to decide the granularity of clusters in each level during the agglomerative procedure. Label propagation and modularity-based algorithms are interpretable and intuitive for flat clustering, rather than for deciding number of levels. The level numbers of LOUVAIN and HLP are simply obtained by the number of rounds of recursions, which is much less interpretable. As shown in the experiments on HSBM models, these two algorithms are easy to miss ground truth levels. In contrast, our framework analyzes the variations of costs when compress the overlength path on cluster trees. Intuitively, a tree edge on a sparse level, which connects an internal node, say $v$, of low conductance to its parent, is less likely to be compressed than those not on sparse levels, because the cost will be much damaged when too many edges (that treat $v$ as LCA) in the original graph enhance their LCA to $v$’s parent.
>
> For the unclear descriptions, we have tried our best to restate most of them. For Lines 268-270, it is hard to give further explanation on local information measurement in a limited space. Consider a local random walk conditioned on the $u$-triangle which corresponds to a subgraph induced by the children of $u$. Stratifying this subgraph is essentially a relatively independent clustering task. From the information-theoretic perspective, the uncertainty of the random walk on this subgraph can be measured locally by structural entropy. This intuition helps us build the local clustering structure in this $u$-triangle.
>
> For the questions,
>
> (1) Sorry for the negligence in the definition of $g$. It should be “a general function $g(|L|,|R|)$, where $L$ and $R$ are the two children of $u\vee v$, respectively.” We have corrected it in the submission.
>
> (2) For the ultrametric, it is the property of isosceles triangles with longer equal sides (See Def. 2.1, [8]). The authors in [8] defined similarity and dissimilarity generated from minimal ultrametric to formulate generating trees there (Def. 2.2, [8]), and in turn defined the admissible cost function (Def. 3.1, [8]). Concretely, a cost is admissible, if for all similarity-based graphs generated from a minimal ultrametric. A cluster tree achieves the minimum cost if and only if it is a generating tree. Due to the space limit, we didn’t explain these concepts in our submission.
>
> (3) Sorry for our negligence. Line 10 in Algorithm 3 should not be indented. We have corrected it.
>
> (4) $\Delta_t(\mathcal{H})$ is the variation of $\delta_t(\mathcal{H})$ that is the amount of structural entropy reduced by the $t$-th round of stratification. $\Delta\mathcal{H}(u)$ is the structural entropy reduced by stratifying the $u$-triangle. So, $\delta_t(\mathcal{H})$ is in fact the sum of $\Delta\mathcal{H}(u)$ over all the internal nodes $u$ that are on the sparsest level.
>
> (5) Sorry for a typo in the definition of $\Delta_t\mathcal{H}$ for which the two terms were reserved. It should be $\delta_{t-1}(\mathcal{H})-\delta_t(\mathcal{H})$. I guess this typo puzzled the reviewer a lot.

---

> > ### Comment · Reviewer_xKCS · 2022-08-05
> > **Thanks and Follow-Up**
> >
> > I thank the authors for their extensive response.
> >
> > I still have a different opinion regarding the interpretability. The method itself and the cost function are not very easy to comprehend (at least not in the current presentation), which severly hampers interpretability. But this is an issue that I would like to discuss best with other reviewers, and I am willing to change my mind (and score) if the other reviewers found the approach interpretable.
> >
> > I also have further concerns about the experimental setup. I believe the authors have misunderstood my comment regarding cost(SE). If cost(SE) is an inadequate measure for evaluating the approach (as the authors seem to argue now), then why is it used? If it is an adequate measure (and it seems so, since the SE in cost(SE) is the same SE as in HCSE, if I understood correctly), then why does HCSE not achieve the best scores?
> >
> > Finally, the authors have argued that their approach is particularly useful for hierarchical structures. If this is so, I think it is important to compare the approach to graph clustering approaches for this problem setting, such as https://arxiv.org/abs/1010.0431.

---

> > > ### Author Response · Authors · 2022-08-07
> > > **Thank the reviewer and response**
> > >
> > > Thank the reviewer for the response.
> > >
> > > For the interpretability, we would like to emphasize that our framework can be collaborated with any cost function, and it is an interpretable process in the sense that it stratifies the sparsest levels one by one, in contrast to the HAC fashion. So, the cost function itself that guides the stretch and compress operations is not the key point in the interpretability of the framework. It is an inadequate measure for evaluating the non-binary cluster tree, but in fact, we do not yet have any reasonable cost for non-binary clustering, since minimizing any cost function will lead to binary clustering. However, we can compare them with certain hierarchical numbers.
> > >
> > > Therefore, for the second concern about cost(SE) used in the experiments, we list in the paper the cost values of both cost(SE) and cost(Das) when we use SE (essentially cost(SE), you are correct) in the framework to illustrate that the results are at least as good as LOUVAIN and HLP (all for cost(Das) and some for cost(SE)) when the hierarchical numbers are similar.
> > >
> > > Further investigation in non-binary clustering with the problem settings used in Rosvall and Bergstrom’s work is a valuable suggestion. Thank the reviewer again!

---

> > > > ### Comment · Reviewer_xKCS · 2022-08-08
> > > > **Thanks**
> > > >
> > > > Thanks for the explanations. I agree that the algorithmic aspects are somewhat interpretable. But the statement that the cost can be simply exchanged by some other cost bothers me a bit: If this is done, then how are the first and second part of the paper connected? (I think a similar concern was already raised by another reviewer.) If cost(SE) is not essential, then I would suggest performing evaluations also with other cost functions, and possibly shorten the first part of the manuscript.
> > > >
> > > > In summary, I see great potential in the paper, but currently I am not convinced by the presentation. I will stick with my current score for now.

---

### Official Review · Reviewer_BgEL · 2022-07-11

**Rating:** 7
**Confidence:** 3
**Soundness:** 3 good
**Presentation:** 3 good
**Contribution:** 3 good

**Summary:**

This paper studies the hierarchical clustering problem on graphs. This paper introduces a simple new objective function for measuring the quality of a hierarchical clustering tree which is in some way more 'natural' than the one proposed by Dasgupta. In particular, the new objective function encourages the tree to be balanced.

The paper also provides an algorithm, closely related to the one by Manghiuc and Sun [15], and proves that it achieves an O(1)-approximation of the optimal objective for well-clustered graphs.

Finally, the paper describes a heuristic algorithm for hierarchical clustering which is neither bottom-up nor top-down and demonstrates that this new clustering perspective can produce good results in practice.

**Questions:**

I would be interested in a little more discussion on the implications of Theorem 3.1. Given that *any* tree gives an $O(1/\Phi)$ approximation, does this mean that an $O(1/\Phi)$-approximation (e.g. the guarantee on Line 71) is somehow 'nothing special' and doesn't really tell us anything about the resulting tree?

### Minor issues
Minor issues which do not affect my evaluation:
* Line 31 - when LCA is introduced it seems to mean the *node* which is the LCA of $u$ and $v$, however throughout the paper it is used to mean the *subtree* rooted at the lowest common ancestor
* Lines 36-37 - I didn't understand this phrase: "a 'natural' ground truth tree in an axiomatic sense therein". Similarly, the sentence on Lines 41-43 is not clear.
* Line 49-50 - strange phrasing - maybe should be "An optimal cluster tree whose height is $\log(n)$ is intuitively..."
* Line 73 typo: 'rencent' -> 'recent'
* Line 127: should this be "$G[S]$ the subgraph *induced* by $S$"?
* Line 143: if $u$ and $v$ are vertices, then the notation $|u|$ in $g(|u|, |v|)$ has not been defined. Similarly the phrase "of their numbers" on line 152 is not precise.
* Line 182 - should this be "the term $\log(\mathrm{vol}(u \lor v))$"?
* Line 186 typo: "Theorem 2.1" -> "Proposition 2.1".
* Theorem 3.1 - $\mathrm{OPT}$ has not been defined.
* Line 208 typo: "there" -> "their"
* Line 353 grammar: "Whether it is a necessary condition?" needs revision



**Limitations:**

The authors adequately address the limitations of the paper.

**Strengths And Weaknesses:**

### Strengths
This paper makes a valuable contribution to an important problem. The newly introduced objective function is natural, and well motivated. It is useful to have an alternative objective function to the one proposed by Dasgupta, and this paper provides some nice discussion on the relative merits of the two objectives. The paper is generally well written and easy to follow.

### Weaknesses
The primary weakness of the paper is that the proposed practical algorithm, and the one used in the experimental evaluation, does not have any theoretical guarantee on its performance.

---

> ### Author Response · Authors · 2022-08-02
> **Response to the concerns and questions.**
>
> Thank the reviewer for the insightful comments. For the weakness and questions, our responses follow.
>
> For the theoretical guarantee on the practical algorithm, we emphasize that, as our best knowledge, there is no proper cost function for the non-binary clustering problem (as we discussed at the end of Section 5 and future discussions). So we even don’t know the objective to approximate. The purpose of our experiments on real datasets is to verify that our explainable framework for non-binary clustering can generate quality cluster trees. Our algorithm HCSE achieves competitive costs over the popular non-binary hierarchical clustering algorithms LOUVAIN and HLP while maintaining reasonable number of hierarchies. Cost function for non-binary clustering is still an significant open problem.
>
> For the question about the $O(\Phi(G)^{-1})$-approximation, yes, an $O(\Phi(G)^{-1})$-approximation is somehow 'nothing special' and doesn't really tell us anything about the resulting tree. Furthermore, we can observe that the ratio of upper and lower bounds of both cost(SE) (with minimum edge weight at least 1) and the original structural entropy defined in Line 136 (with any scale of weights) ranges from 1 to $\log vol(V)$. Therefore, any cluster tree for any graph gives a $\log vol(V)$-approximation. When $vol(V)=poly(n)$ ($n$ is the graph size), it is an $O(\log n)$-approximation. In this case, when $\Phi(G)=o(1/\log n)$, we can always keep this guarantee at $O(\log n)$.
>
> For the minor issues, we response as follows:
>
> (1) We use LCA to indicate an internal node on cluster tree, but sometimes without ambiguity, we mean the vertex set in the original graph that corresponds to this LCA.
>
> (2) The authors in [8] proposed axiomatic conditions for the cost functions to evaluate the hierarchical clustering trees. They defined admissible cost function that satisfies these conditions. Concretely, a cost is admissible, if for all similarity-based graphs generated from a minimal ultrametric, a cluster tree achieves the minimum cost if and only if it is a generating tree that is a “natural” ground truth tree in an axiomatic sense. An important condition is that any binary cluster tree for a clique should have equal cost. Due to the space limit, we didn’t explain these concepts in our submission.
>
> (3) We have restated this sentence as “At least, an optimal cluster tree whose height is logarithm of graph size $n$ is intuitively more reasonable than a caterpillar shaped cluster tree whose height is $n-1$.”
>
> (4) Revised.
>
> (5) Revised.
>
> (6) There is a negligence in defining function $g$, which should be the sum of the sizes of $u\vee v$’s two children. We have corrected it.
>
> (7-11) Revised.

---

### Official Review · Reviewer_cybk · 2022-07-12

**Rating:** 3
**Confidence:** 3
**Soundness:** 2 fair
**Presentation:** 1 poor
**Contribution:** 2 fair

**Summary:**

The paper proposes an information-theoretic cost function for hierarchical clustering of graphs. The cost function is inspired by structural information theory, which measures the complexity of a hierarchical network with structural entropy. The paper proves that minimizing the structural entropy cost function is equivalent to minimizing a cost function similar to admissible functions defined in previous works. The paper proposes two constant factor approximation algorithms for expander-like and for well-clustered graphs. It also describes a new framework for non-binary hierarchical clustering (i.e., internal nodes may have >2 children), which can be used with structural entropy cost or other cost functions. The non-binary hierarchical clustering approach is tested on simulated and real data, and compared with previously proposed methods LOUVAIN and HLP.

**Questions:**

What is a real-world example where it would make sense to cluster a clique into two balanced clusters instead than in arbitrary clusters?

Is there a reason why the binary clustering algorithms have not been experimentally evaluated?

Lemma 3.1: what is \Phi_G(P_i)? It is not defined in the paper.

Theorem 3.2: if \alpha = O(k^6 \sqrt{\lambda_{k-1}}), it seems that the approximation factor can be negative

Line 249: “while maintaining each intra-link and ignoring each internal edge of v_i” isn’t this a contradiction?

Section 5: using cost(Das) and cost(SE) seems to be biased to favor your algorithm, since it does use cost(SE) and cost(Das) is somehow related to cost(SE) by Proposition 2.1. Is this not correct?

Fig.1: according to the definition of inflection point in section 4, t=4 is not an inflection point for the blue plot (\delta_4 is negative, while \delta_5 is essentially 0). What is the inflection point?

Section 5, “real datasets”: what is the message of such results?


**Limitations:**

There is no discussion, but it seems not needed for such a paper.

**Strengths And Weaknesses:**

Strengths:
- the proposed approach for non-binary clustering builds on a nice idea
- the proved relation between structural entropy cost and a cost analogous to admissible functions (Proposition 2.1) is a nice result bridging the information-theoretic and combinatorial approaches

Weaknesses:
- the method works for graph data, but this is not clear in the title and abstract
- the motivation in the Introduction is not very clear, since the shortcomings and weaknesses of admissible cost functions are not well described. The main motivation seems to be that when a clique is to be split in clusters, the authors suggest that it is better to split it in balanced clusters, while admissible cost functions would split the clique arbitrarily. However, the latter choice seems more natural to me, since from the point of view of clustering all partitions of a clique are equivalent.
- The paper presents very different contributions, with Section 4 (the algorithm for non-binary clustering) that does not seem to belong in the “information-theoretic perspective” on hierarchical clustering (it is a generic framework, not tied to the information-theoretic perspective of the previous sections)
- The evaluation is performed only for the non-binary clustering algorithm, making the experimental evaluation less significant
- The experimental evaluation is not entirely clear. In particular, the procedure to choose the number of levels does not seem to match with the plots
- In section 2 v is used to represent both a node of T and the set of nodes of G that v represents, which is sometimes confusing (see below for additional requests of clarification)
- (minor) Some figures appear only the supplement but they are references in the main text without marking them as Supplemental Figures or as appearing in Supplemental Material

---

> ### Author Response · Authors · 2022-08-02
> **Response to the concerns and questions.**
>
> Thank the reviewer for the insightful comments. For the weakness and questions, our responses follow.
>
> For the applicability of our method, for data points embedded in some vector space, a graph based on some metric can be constructed easily. Graph and vector data are generally dual in the sense that they can converted to each other properly by some commonly used methods like graph embedding and similarity metrics, etc. This viewpoint has also been implicit in the study line of hierarchical clustering, e.g., in [10], [16], [15]. All the cost functions are also fit for both scenarios. Although we didn't clarify this in the title and abstract, the inputs of our algorithms mentioned in the abstract are clarified as graphs.
>
> For the motivation of our study, we provide an information-theoretic perspective for the hierarchical clustering and bridge it with the widely used combinatorial definitions based on Dasgupta’s cost function. The balance of cluster trees is an important topic in clustering. It has many applications in practice, for example, load balancing for cloud computing and management, network design, and avoidance of outlier cluster formation, etc. Our cost function addresses this issue naturally. In contrast, Dasgupta’s cost and its derivatives didn’t consider the balance at all. Our cost is in fact a trade-off between the two factors: connection and balance. When connections become uniform (in cliques), the factor of balance determines the optima of cluster trees. We do not mean that splitting the clique arbitrarily is not natural. It is natural when consider only connections, but it’s imperfect since the balance factor has been ignored. At least, it’s disputable to take it as an axiomatic condition in [8].
>
> For the difference of contributions, we clarify the relationship of our contributions. Firstly, we propose a new objective function from the information-theoretic perspective, and then for this new cost, we give approximation algorithms in two classic scenarios. This is a traditional route of research for a newly defined problem in algorithmic study. The approximation algorithms are major contributions, and we didn’t focus on the performance of other clustering algorithms when coupled with our cost function. This is also the reason why we didn’t evaluate our binary algorithm and compare it with other binary ones experimentally (Weakness 4 and Question 2). Secondly, we propose an explainable framework for non-binary clustering which is practically useful. The information-theoretic perspective is certainly suitable for non-binary clustering, and our cost function can be incorporated in this framework. More information will be given in the answers to Questions 6 and 8.
>
> For the questions,
>
> (1) A real-world example seems hard to give. Let’s consider a weighted clique of size $n$ with all edge weights equal to 1, except $n-1$ edges associated with a single vertex $v$ have weights $1-\epsilon$. To minimize cost(Das), this single vertex $v$ must be divided from other $n-1$ ones first and $\{v\}$ is a cluster of singleton at a very high level. Comparatively, for our cost function, according to the discussion at the end of Section 2, such slight variations on edge weights will not influence the optimal cluster tree wildly and $v$’s neighbors should be divided into two equal groups at the first level, due to the holdback force of balance.
>
> (2) As stated above, evaluation of different algorithms for binary cluster trees under our new cost function or any other objective is not our purpose. We propose approximation algorithms as theoretical results for the new cost and focus experiments on non-binary clustering.
>
> (3) $\Phi_G(P_i)$ denotes the conductance of $P_i$ in graph $G$. We have claimed it before Lemma 3.1.
>
> (4) In Theorem 3.2, $k\geq 1$, $\alpha\in (0,1)$, $\beta>0$. So the approximation guarantee is always positive.
>
> (5) Sorry for the typo in Line 249, “intra-link” should be “inter-link”. We have corrected it.
>
> (6) cost(SE) and cost(Das) have the same form of $\sum_{(u,v)\in E} w(u,v) g(u,v)$ and share the same intuition. There is not too much bias to favor our framework when they are used. It is indeed better to compare with other costs for stronger robustness of our framework. But we need to emphasize that such a comparison is not our purpose. We just verify that our explainable framework for non-binary clustering can generate quality cluster trees that are competitive with what the popular LOUVAIN and HLP output.
>
> (7) Sorry for a typo here, the definition of $\Delta_t\mathcal{H}$ was reserved. It should be $\delta_{t-1}(\mathcal{H})-\delta_t(\mathcal{H})$. I guess this typo puzzled the reviewer a lot (Weakness 5).
>
> (8) We verify that our explainable framework for non-binary clustering can generate quality cluster trees. Our algorithm HCSE achieves competitive costs over the popular non-binary hierarchical clustering algorithms LOUVAIN and HLP while maintaining reasonable number of hierarchies.

---

### Meta-Review · Area_Chair_G5k8 · 2022-08-24

**Recommendation:** Reject
**Confidence:** Certain

**Metareview:**

The paper introduces a new cost function inspired by structural information theory for hierarchical clustering of graphs. The paper show the relationship of the new cost function with previous work and proposes two algorithms to obtain a constant factor approximation for  expander-like and for well-clustered graphs.

The paper presents some new ideas and some interesting results although additional work is needed and the paper is not ready for publication in the current state. The main weaknesses highlighted by the committee are:
- the new cost function is not well-motivated and additional discussion will be needed
- the experiments are not too convincing
- the current presentation should be substantially improved before acceptance

**Award:**

No

---

### Decision · Program_Chairs · 2022-09-14

Reject